# Efficient Neural Controlled Differential Equations via Attentive Kernel Smoothing

Egor Serov [1]    Ilya Kuleshov [1]    Alexey Zaytsev [1 2]

## Abstract

Neural Controlled Differential Equations (Neural CDEs) provide a powerful continuous-time framework for sequence modeling, yet the roughness of the driving control path often restricts their efficiency. Standard splines introduce high-frequency variations that force adaptive solvers to take excessively small steps, driving up the Number of Function Evaluations (NFE). We propose a novel approach to Neural CDE path construction that replaces exact interpolation with Kernel and Gaussian Process (GP) smoothing, enabling explicit control over trajectory regularity. To recover details lost during smoothing, we propose an attention-based Multi-View CDE (MV-CDE) and its convolutional extension (MVC-CDE), which employ learnable queries to inform path reconstruction. This framework allows the model to distribute representational capacity across multiple trajectories, each capturing distinct temporal patterns. Empirical results demonstrate that our method, MVC-CDE with GP, achieves state-of-the-art accuracy while significantly reducing NFEs and total inference time compared to spline-based baselines.

## 1. Introduction

Sequential data processing is a fundamental challenge across a wide range of disciplines, including the analysis of medical records (Harutyunyan et al., 2019), financial dynamics (Zhang et al., 2018), and physical systems (Raissi et al., 2019). Most neural architectures for sequential data implicitly assume that observations lie on a regular temporal grid, enabling the direct application of discrete-time models such as recurrent or convolutional networks. However, this assumption is frequently violated in real-world settings.

Sensor failures, asynchronous measurements, and missing data often result in irregularly sampled time series.

A more rigorous approach is to treat the latent state evolution as a continuous-time process. This paradigm shift gained traction with Neural ODEs (Chen et al., 2018), which model the dynamics of the hidden state via ordinary differential equations. However, standard Neural ODEs are determined by their initial state and do not naturally adjust to incoming observations over time. To address this, Neural Controlled Differential Equations (Neural CDEs) (Kidger et al., 2020) were introduced as a principled framework for processing irregularly observed sequences. Their central idea is to represent the input data as a continuous-time control signal $\mathbf{X}(t)$, obtained via interpolation, and to evolve a hidden state $\mathbf{z}(\mathbf{t})$ according to the controlled differential equation

$$\frac{\mathrm{d}\mathbf{z}(t)}{\mathrm{d}t} = f_\theta(\mathbf{z})\frac{\mathrm{d}\mathbf{X}(t)}{\mathrm{d}t}. \tag{1}$$

where the vector field $f_\theta$ is parameterized by a neural network. This formulation allows Neural CDEs to naturally handle missing values and irregular sampling while maintaining a continuous-time latent representation.

The computational cost of Neural CDEs is dominated by the numerical integration of this equation. In particular, it is closely tied to the number of times the solver evaluates the vector field function $f_\theta$, commonly referred to as the NFE. Modern implementations typically rely on adaptive-step solvers, which aim to minimize NFE while satisfying a prescribed error tolerance by taking larger steps in regions where the dynamics are smooth.

However, there is a direct trade-off: NFE is highly sensitive to the regularity of the input trajectory. In practice, real-world time series are often noisy, and accurate interpolations of such data can produce highly irregular or jagged paths. This forces adaptive solvers to take many small integration steps, dramatically increasing NFE and computational cost. As a consequence, Neural CDEs can be an order of magnitude slower than more traditional time-series modeling approaches, limiting their practical applicability.

In this work, we propose replacing precise interpolation schemes with smoothing-based alternatives, thereby substantially reducing trajectory irregularity and accelerating

[1]Applied AI Institute [2]Risk AI Research Lab. Correspondence to: Egor Serov <e.serov@applied-ai.ru>.

*Proceedings of the 43rd International Conference on Machine Learning*, Seoul, South Korea. PMLR 306, 2026. Copyright 2026 by the author(s).

numerical integration by up to an order of magnitude, without sacrificing predictive performance. To mitigate the potential loss of high-frequency information caused by over-smoothing, we introduce an attention-based architecture inspired by Q-Former models (Li et al., 2023a). This mechanism adaptively aggregates information from multiple smoothed versions of the input, each capturing different temporal scales or regions of the sequence.

Conceptually, our approach can be viewed as a novel form of temporal parallelization for Neural CDEs: rather than integrating a single high-fidelity trajectory, we distribute interpolation precision across several smoother Neural CDE paths and learn to recombine them. This yields significant computational savings while preserving the expressive power of the continuous-time model.

Our main contributions are as follows:

- We identify stiffness as a primary driver of the high NFE in Neural CDEs, and empirically demonstrate that accurate interpolations of noisy real-world data can severely degrade solver efficiency.

- We propose replacing precise interpolation schemes with smoothing-based Kernel and GP alternatives for Neural CDEs. Such smoothing significantly reduces trajectory roughness and accelerates numerical integration by up to an order of magnitude.

- To counteract information loss induced by over-smoothing, we introduce an attention-based architecture inspired by Q-Former models named Multi-View Controlled Differential Equations (MV-CDE) and its convolutional extension, MVC-CDE. They adaptively aggregate information from multiple smoothed versions of the input, each capturing different temporal regions or scales.

- We validate our approach on real-world classification benchmarks. Our method achieves state-of-the-art accuracy—performing competitively alongside recent linear-time state-space models such as Mamba—while demonstrating speedups of $5\times$ to $14\times$ over standard Neural CDEs, along with superior robustness to additive noise.

**Conflict of Interest Disclosure.** The authors declare that there are no financial conflicts of interest related to this work.

## 2. Related Work

**Neural Controlled Differential Equations.** Neural ODEs (Chen et al., 2018) introduced continuous-depth modeling, yet their dependence on fixed initial conditions limits them to tabular data or requires hybrid discrete-continuous updates like ODE-RNNs (Rubanova et al., 2019). To address this, Kidger et al. (2020) formulated Neural CDEs, in which the dynamics are driven by a continuous control path interpolated from observations. While this architecture effectively generalizes Recurrent Neural Networks (Oh et al., 2025) to irregular time series, a fundamental limitation remains: the computational cost is intrinsically tied to the regularity of the trajectory. Regardless of the learned vector field, the solver must traverse the geometric complexity of the driving path.

**Interpolation and Path Regularity.** Significant efforts have focused on regularizing the latent dynamics to reduce NFEs, employing penalties on higher-order derivatives (Kelly et al., 2020), optimal transport (Finlay et al., 2020), or stochastic end-time regularization (Ghosh et al., 2020). However, since Neural CDE dynamics are multiplicative, the solver step size is still determined by the roughness of the driving control path. Standard exact interpolation schemes force the path through every noisy observation, resulting in high-frequency variations and large derivatives (de Boor, 1978; Reinsch, 1967). As the local truncation error scales with these derivatives (Hairer et al., 1993), adaptive solvers must drastically reduce step sizes to maintain accuracy (Dormand & Prince, 1980; Morrill et al., 2022). Unlike prior work confined to deterministic interpolants, we propose explicit path smoothing via non-parametric kernels (Wand & Jones, 1995) or Heteroscedastic Gaussian Processes (Rasmussen & Williams, 2006) to decouple integration cost from input noise.

**Stochastic and Probabilistic Approaches.** Incorporating uncertainty directly into differential equations, Neural SDEs (Li et al., 2020; Kidger et al., 2021b) extend the CDE framework with diffusion terms. While powerful, this necessitates complex stochastic integrators like Euler-Maruyama (Kloeden & Platen, 1989), significantly increasing computational cost. Probabilistic frameworks, such as Neural Processes (Garnelo et al., 2018) and Spatio-Temporal Point Processes (Chen et al., 2021), model data as unordered sets or discrete event intensities. Consequently, they either lack the explicit causal structure of recurrent systems or do not track the continuous evolution of signal values. Other lines of research address irregular sampling by employing Gaussian Processes as feature extractors or adapters before passing data to RNNs (Li & Marlin, 2016; Futoma et al., 2017), or by utilizing continuous-time interpolation and attention networks (Shukla & Marlin, 2019; 2020). While effective, these models operate primarily in discrete time after the initial adaptation phase. In contrast to introducing expensive stochastic latent dynamics or reverting to discrete RNNs, we use Gaussian Processes solely as a robust smoothing mechanism—relying on the optimal parameter estimation properties of GPs (Zaytsev & Burnaev, 2014) to construct

regularized continuous control paths for Neural CDEs.

**Efficient Integration and Structured Approaches.** Algebraic approaches aim to optimize integration by summarizing input data over intervals. Neural Rough Differential Equations (Morrill et al., 2021) and Log-Neural CDEs (Walker et al., 2024) utilize signatures (Lyons, 1998; Chevyrev & Kormilitzin, 2026; Lyons & McLeod, 2025) to effectively decouple the step size from the raw sampling rate. Other works focus on solver mechanics, such as relaxing error tolerances via seminorms (Kidger et al., 2021a), or architectural efficiency: Structured Linear CDEs (Walker et al., 2025) employ block-diagonal state transitions to accelerate vector field evaluations. We note that our Multi-View architecture shares this block-diagonal structure, but uses it to integrate multiple trajectory views simultaneously rather than for parallel associative scans. Finally, Neural Flows (Bilos et al., 2021) bypass iterative integration entirely by modeling the solution map directly. However, this eliminates the explicit causal structure necessary for adapting to evolving input streams. Unlike these methods, which focus on algebraic compression or bypassing the solver, our work targets the geometric irregularity of the input path itself as the primary source of numerical stiffness. A parallel line of research, Mechanistic Neural Networks (Pervez et al., 2024), targets integration efficiency by reformulating linear ODE systems as constrained optimization problems and solving them using parallelized differentiable linear-programming layers. While highly efficient for linear scientific modeling, such approaches cannot be easily extended to general non-linear Neural CDE vector fields. In contrast, our approach is complementary: rather than modifying the solver's internal algebraic structure, we optimize the geometric smoothness of the driving input path itself, thereby directly mitigating stiffness for general-purpose adaptive non-linear solvers.

**Latent Path Learning and Attention Mechanisms.** While discrete transformers (Zhou et al., 2021; Wu et al., 2021) address long-term dependencies on fixed grids, continuous attention mechanisms—such as ContiFormer (Chen et al., 2023), Attentive NCDEs (Jhin et al., 2024), and MA-NODE (Havaei et al., 2025) adapt this to irregular data. However, these methods primarily use attention to capture historical correlations or fuse multi-scale dynamics, often increasing computational cost. Similarly, generative frameworks like EXIT-NCDE (Jhin et al., 2022), Neural Lad (Li et al., 2023b), and stability-focused like DeNOTS (Kuleshov et al., 2026) prioritize expressivity or extrapolation, frequently at the expense of higher NFEs. In contrast, we employ attention not for temporal dependency modeling, but as a reconstruction mechanism. Inspired by Q-Former (Li et al., 2023a), our Multi-View architecture uses learnable queries to recover high-frequency information from explicitly smoothed control paths. While Log-Neural CDEs (Walker et al., 2024) address efficiency via algebraic sum-

marization, our work establishes a parallel geometric alternative, demonstrating that path smoothing combined with attentive recovery offers a robust solution to the solver bottleneck.

**Summary and Baselines.** To rigorously evaluate our contribution, we select next baselines: GRU-D (Che et al., 2016), ODE-RNN (Rubanova et al., 2019) , Linear and Cubic Neural CDEs (Kidger et al., 2020). Additionally, we include Log-NCDE (Walker et al., 2024) and the recent linear-time state-space model Mamba (Gu & Dao, 2024) as primary benchmarks for computational efficiency. The critical research gap we address is the high sensitivity of adaptive solvers to input noise, which remains a bottleneck for standard Neural CDEs and is only indirectly addressed by algebraic signature-based methods. Our novelty lies in a decoupling of the integration cost from signal noise via smoothing, while recovering lost details through a multi-view attentive mechanism.

## 3. Method

### 3.1. Problem Formulation

We consider the task of supervised learning on time series. Let $\mathcal{S} = \{(t_k, \mathbf{x}_k)\}_{k=1}^N$ be an input sequence, where $t_k \in \mathbb{R}$ represent strictly increasing time stamps and $\mathbf{x}_k \in \mathbb{R}^d$ are observed feature vectors. The sampling rate is characterized by the time intervals $\Delta t_k = t_{k+1} - t_k$. It reduces to a constant $\Delta t$ for regular series. The objective is to learn a mapping from $\mathcal{S}$ to a target $y$. For simplicity, we assume that $t_1 = 0$ and $t_N = T$. The sequence is processed to obtain a latent embedding vector, which is subsequently passed to a linear classification or regression model to produce a prediction $\hat{y}$.

### 3.2. Neural Controlled Differential Equations

Neural CDE (Kidger et al., 2020) models the latent state $\mathbf{z}(t)$ as the solution to a differential equation driven by a continuous control path $\mathbf{X}(t) : [0, T] \to \mathbb{R}^d$, which is constructed by interpolating the discrete observations $\mathcal{S}$. The dynamics are governed by:

$$\mathbf{z}(T) = \mathbf{z}(0) + \int_0^T f_\theta(\mathbf{z}(t)) \frac{\mathrm{d}\mathbf{X}(t)}{\mathrm{d}t} \mathrm{d}t, \qquad (2)$$

where $f_\theta$ is a neural network parameterizing the vector field. The initial state $\mathbf{z}(0)$ is typically a learned linear projection of $X(0)$. In practice, this integral is computed by an adaptive ODE solver (such as Dormand-Prince (Dormand & Prince, 1980; Hairer et al., 1993)) applied to the equivalent ODE system (1). These solvers dynamically adapt the step size $\eta_i = t_{i+1} - t_i$ to keep the Local Truncation Error (LTE) below a tolerance $\delta$. Throughout our analysis, we formally define the Number of Function Evaluations (NFE) as the

total number of calls to the underlying vector field $f_\theta$ during this adaptive integration process, including both accepted and rejected steps. To analyze the computational complexity, we first establish the relationship between control-path regularity and the solver step size.

**Theorem 3.1** (NFE dependence on Control Path). *For an adaptive ODE solver (e.g., Dormand-Prince) of order $p$ with error tolerance $\delta$, the total NFE is bounded by the integral of the inverse step size up to a positive constant:*

$$NFE \lesssim \int_0^T \left( \frac{1}{\delta} \left\| \frac{\mathrm{d}^{p+1}\mathbf{X}(t)}{\mathrm{d}t^{p+1}} \right\| \right)^{\frac{1}{p+1}} \mathrm{d}t.$$

This theorem establishes that NFE is strictly determined by the $L_{1/(p+1)}$-quasi-norm of the $(p+1)$-th derivative of the control path.

### 3.3. Limitations of Deterministic Splines

Standard implementations construct $\mathbf{X}(t)$ using natural cubic splines or linear interpolation. While these methods guarantee $\mathbf{X}(t_k) = \mathbf{x}_k$, exact interpolation rigidly couples path regularity with input variations. Applying Theorem 3.1, we derive explicit complexity estimates over the fixed size interval $[0, T]$.

For linear and cubic splines, the integration cost scales with the first and second finite differences, respectively:

$$\text{NFE}_{\text{lin}} \lesssim \sum_k \|\Delta\mathbf{x}_k\|, \quad \text{NFE}_{\text{cub}} \lesssim \frac{1}{\Delta t} \sum_k \|\Delta^2\mathbf{x}_k\|,$$

where $\Delta\mathbf{x}_k$ denotes the discrete derivative. In both cases, path roughness is determined by the input sequence. The solver is forced to traverse the exact geometry of the raw noisy observations, leading to a high NFE. See Figure 1 for a visualization of solver step distribution in standard splines.

### 3.4. Single Path Smoothing

To address the stiffness inherent to spline interpolation, we propose methods that decouple the trajectory smoothness

from the input data. We first introduce Kernel Smoothing as a baseline to motivate the decoupling, then present Gaussian Process Smoothing, which constitutes our contribution.

**Kernel Smoothing.** We employ the Nadaraya-Watson estimator with a Gaussian Radial Basis Function (RBF) kernel $k_h(t, t') = \exp\left(-\frac{(t-t')^2}{2h^2}\right)$ to construct a smooth path $X_{kernel}(t)$. Given a bandwidth parameter $h$, the path is defined as:

$$\mathbf{X}_{kernel}(t) = \frac{\sum_{k=1}^N k_h(t - t_k)\mathbf{x}_k}{\sum_{j=1}^N k_h(t - t_j)}.$$

**Gaussian Process (GP) Smoothing.** Alternatively, we place a GP prior on the underlying signal, $\mathbf{X}_{GP}(t) \sim \mathcal{GP}(0, k(t, t'))$. We utilize the same kernel defined above. The control path $\mathbf{X}_{GP}(t)$ is the posterior mean function conditioned on the observations $\mathcal{S}$ with assumed Gaussian noise variance $\sigma^2$. It is computed as:

$$\mathbf{X}_{GP}(t) = \mathbf{k}_h(t)^\top (\mathbf{K}_h + \sigma^2\mathbf{I})^{-1}\mathbf{X},$$

where $\mathbf{X} \in \mathbb{R}^{N\times d}$ is the matrix of stacked feature vectors. The kernel vector $\mathbf{k}_h(t)$ and Gram matrix $\mathbf{K}_h$ are defined as in standard GP regression.

While exact GP inference has cubic complexity $\mathcal{O}(N^3)$ with respect to sequence length due to matrix inversion, this is a static, one-time preprocessing cost. This initialization overhead remains negligible compared to the integration efficiency gains. As proven in Appendix A.5, GPs provide the optimal linear reconstruction, justifying the cubic pre-processing cost by ensuring that the smoothed trajectory remains maximally faithful to the underlying signal compared to heuristic kernel estimates.

**Theorem 3.2** (Derivative Bounds for Smoothing Kernels). *Let $\mathbf{X}_h(t)$ be constructed via Nadaraya-Watson regression or as the posterior mean of a GP with a stationary RBF kernel with lengthscale $h$. The $p$-th derivative of the trajectory*

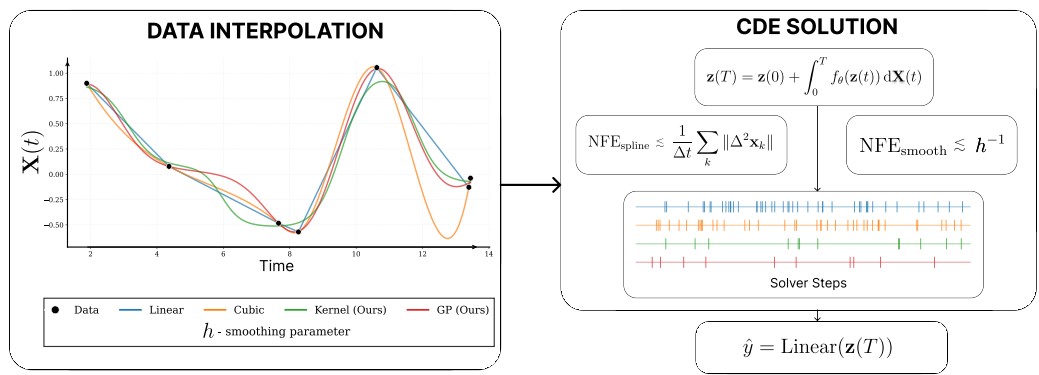

*Figure 1.* Standard Neural CDE pipeline for different types of path construction.

*satisfies the uniform bound:*

$$\sup_{t\in[0,T]}\left\|\frac{\mathrm{d}^p\mathbf{X}_h(t)}{\mathrm{d}t^p}\right\|\leq\frac{C_p}{h^p}\|\mathbf{X}\|_\infty,$$

*where $C_p$ is a constant depending on the kernel shape and $\|\mathbf{X}\|_\infty$ is the bound on observations.*

**Corollary 3.3** (Smoothing NFE Scaling). *Substituting the bound from Theorem 3.2 into the NFE integral (Theorem 3.1) yields:*

$$NFE(\mathbf{X}_h(t))\lesssim\int_0^T\delta^{-\frac{1}{p+1}}\left(h^{-(p+1)}\right)^{\frac{1}{p+1}}\mathrm{d}t\lesssim\delta^{-\frac{1}{p+1}}h^{-1}.$$

Selecting $h$ entails a trade-off between statistical bias and computational efficiency (NFE). While classically optimal bandwidths ($h\propto n^{-1/5}$) minimize reconstruction error, they can induce high-frequency stiffness, thereby inflating NFE. We treat $h$ as a task-aware hyperparameter, further detailed in Appendix A, allowing us to optimize for the downstream prediction task.

### 3.5. Multi-View Architecture

**Overview and Motivation.** Standard smoothing can eliminate high-frequency information that may be relevant to the prediction task. To mitigate information loss, we propose a Multi-View CDE (MV-CDE) architecture that integrates multiple distinct trajectories simultaneously. See the complete pipeline is illustrated in Figure 2. Furthermore, to incorporate local temporal context, we introduce the Multi-View Convolutional CDE (MVC-CDE) extension. Rather than relying on a single smoothed approximation, these frameworks construct a set of paths differentiated by their attention weights and smoothness parameters.

**Feature Extraction and Convolution.** Applying interpolation weights based solely on raw pointwise values may fail to capture complex local temporal dependencies. To address this, we introduce an optional feature-extraction stage. Let $\mathbf{u}_k$ denote a context vector associated with the observation $\mathbf{x}_k$.

In the basic MV-CDE setting, the model computes attention scores from observations, setting $\mathbf{u}_k=\mathbf{x}_k$. For MVC-CDE, we account for temporal neighborhoods with a 1D convolutional network $\Phi_{\text{conv}}$ and fixed kernel size. Following VGG (Simonyan & Zisserman, 2015), small kernels capture local context. The network applies discrete temporal convolutions over $\mathcal{S}$ to produce latent descriptors:

$$\{\mathbf{u}_k\}_{k=1}^N=\Phi_{\text{conv}}(\{\mathbf{x}_k\}_{k=1}^N).$$

Specifically, these descriptors $\mathbf{u}_k$ are used exclusively to determine the attention weights of each observation. The continuous control path $\mathbf{X}^{(m)}(t)$ itself remains a function of the original observations $\mathbf{x}_k$, preserving the interpretability and dimensionality of the input signal.

**Weighted Paths.** To capture multi-scale dynamics without over-smoothing, we introduce a multi-head architecture. Inspired by the Querying Transformer (Q-Former) from vision-language modeling (Li et al., 2023a), we employ a set of learnable queries to extract relevant temporal features. We define $M$ learnable query vectors $\mathbf{Q}=[\mathbf{q}_1,\ldots,\mathbf{q}_M]\in\mathbb{R}^{M\times d_{\text{ctx}}}$, where $d_{\text{ctx}}$ is the dimension of $\mathbf{u}_k$. For each head $m$, we compute attention scores $\alpha_{m,k}$ using the context vectors $\mathbf{u}_k$:

$$\alpha_{m,k}=\text{softmax}_k\left(\frac{\mathbf{q}_m^\top\mathbf{u}_k}{\sqrt{d_{\text{ctx}}}}\right).$$

These attention weights determine the relevance of each observation for the $m$-th trajectory. We construct $M$ distinct continuous paths $\mathbf{X}^{(m)}(t)$, utilizing the chosen smoothing scheme. Therefore, we state Weighted Interpolation schemes applied to the original observations $\mathbf{x}_k$:

**Weighted Kernel Path.** The weights modify the kernel density estimate (Cai, 2001):

$$\mathbf{X}^{(m)}(t)=\frac{\sum_{k=1}^N\alpha_{m,k}k_{h_m}(t-t_k)\mathbf{x}_k}{\sum_{j=1}^N\alpha_{m,j}k_{h_m}(t-t_j)}. \tag{3}$$

**Weighted GP Path.** We introduce heteroscedastic noise into the GP posterior formulation (Lázaro-Gredilla & Titsias,

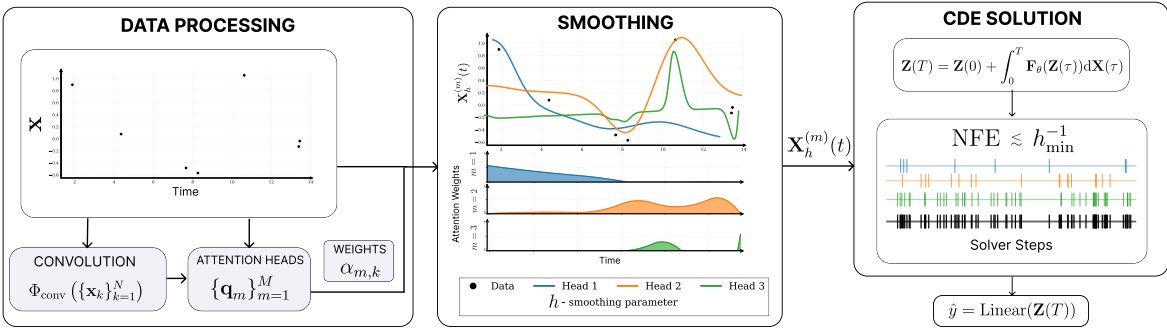

*Figure 2.* Proposed MV-CDE and MVC-CDE architectures.

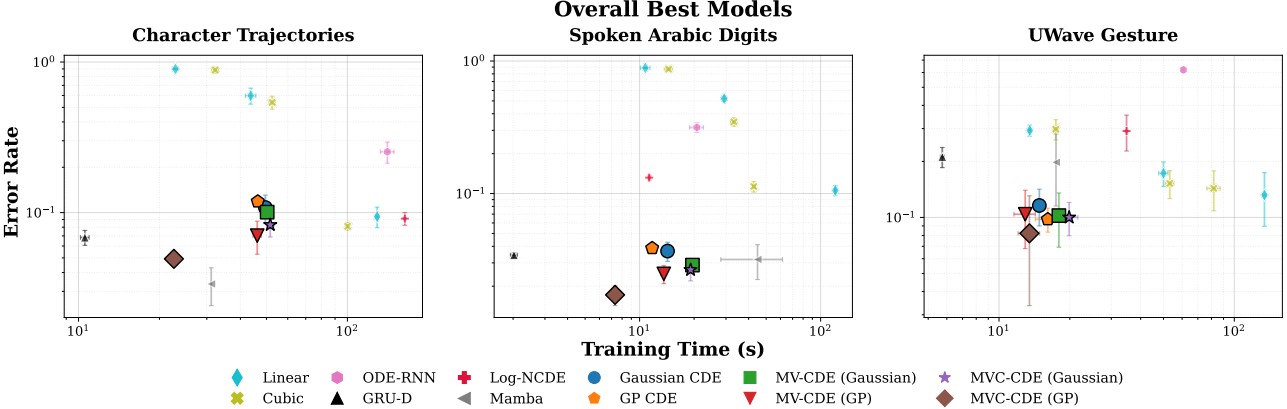

*Figure 3.* **Pareto Efficiency Plot.** Error Rate (log scale) versus Total Training Time (log scale).

2011; Le et al., 2005). We define a diagonal noise covariance matrix $\mathbf{\Sigma}_m$ where the noise variance for the $k$-th observation is inversely proportional to its attention score:

$$(\mathbf{\Sigma}_m)_{kk} = \frac{\sigma_{\text{base}}^2}{\alpha_{m,k} + \epsilon}.$$

Observations with high attention scores are treated as high-precision points with low noise variance, while low-attention points are filtered out as noise.

$$\mathbf{X}^{(m)}(t) = \mathbf{k}_h(t)^\top (\mathbf{K}_h + \mathbf{\Sigma}_m)^{-1}\mathbf{X}, \qquad (4)$$

where the kernel structures $\mathbf{k}_{h_m}$ and $\mathbf{K}_{h_m}$ follow the definitions in Section 3.4, parameterized by head-specific length-scales $h_m$, while $\mathbf{X}$ remains the shared observation matrix.

**Parallel Integration.** The proposed architecture generates a set of $M$ control paths $\{\mathbf{X}^{(m)}(t)\}_{m=1}^M$. These paths are integrated simultaneously within a single ODE solver call. We define the global latent state $\mathbf{Z}(t)$ as the concatenation of the individual head states:

$$\mathbf{Z}(t) = \left[\mathbf{z}^{(1)}(t), \dots, \mathbf{z}^{(M)}(t)\right] \in \mathbb{R}^{M \cdot d_{\text{hidden}}}.$$

Each head evolves according to its own parameterized vector field $f_{\theta_m}$. The combined differential equation is:

$$\frac{d\mathbf{Z}(t)}{dt} = \text{Concat}_{m=1}^M \left( f_{\theta_m}(\mathbf{z}^{(m)}(t)) \frac{d\mathbf{X}^{(m)}(t)}{dt} \right). \quad (5)$$

This approach essentially treats the system as a block-diagonal ODE (Walker et al., 2025), where independent dynamics are solved jointly. It leads to the next theorem:

**Theorem 3.4** (Parallel Integration Bottleneck)**.** *Consider an adaptive ODE solver integrating the block-diagonal system* (5) *with* $\mathbf{Z}(t)$ *with a global error tolerance* $\delta$. *The total NFE scales jointly with the solver tolerance and the minimum smoothing parameter in the ensemble as:*

$$NFE_{total} \lesssim \delta^{-\frac{1}{p+1}} \left( \min_{m=1}^M h_m \right)^{-1}.$$

This formulation highlights a critical design choice. While including heads with very small $h_m$ allows for capturing fine-grained details, it increases the overall computational cost. By optimizing $h_m$, our method balances multi-view representation capability with integration speed. To justify the use of $M > 1$, we observe that single-head models often fail to capture diverse temporal features, while multiple heads provide robustness through ensemble representations.

**Pipeline Summary.** The forward pass begins with an encoding step that maps the input $\mathbf{x}_k$ to $\mathbf{u}_k$ via a CNN or an identity function. This is followed by an attention mechanism that computes $M$-head scores $\alpha_{m,k}$ using learnable queries. These scores are subsequently used in a smoothing phase to generate continuous paths $\mathbf{X}^{(m)}(t)$ through weighted Gaussian processes (4) or kernel regression (3). Finally, the integration step yields the prediction by solving a block-diagonal NCDE driven by the concatenation of these smooth paths.

## 4. Results

In this section, we present the comparative performance of our proposed architectures against the baselines. We primarily focus on classification accuracy and computational efficiency, followed by an ablation study and an analysis of model interpretability.

For detailed experimental settings and dataset descriptions, we refer to Appendix B.

### 4.1. Main Performance Comparison

We evaluated our method across three diverse benchmarks from the UEA Time Series Archive (Bagnall et al., 2018), selected to cover a broad spectrum of signal characteristics. Low-dimensional **CharacterTrajectories** captures high-frequency pen-tip dynamics. High-dimensional **SpokenArabicDigits** represents speech feature coefficients; and

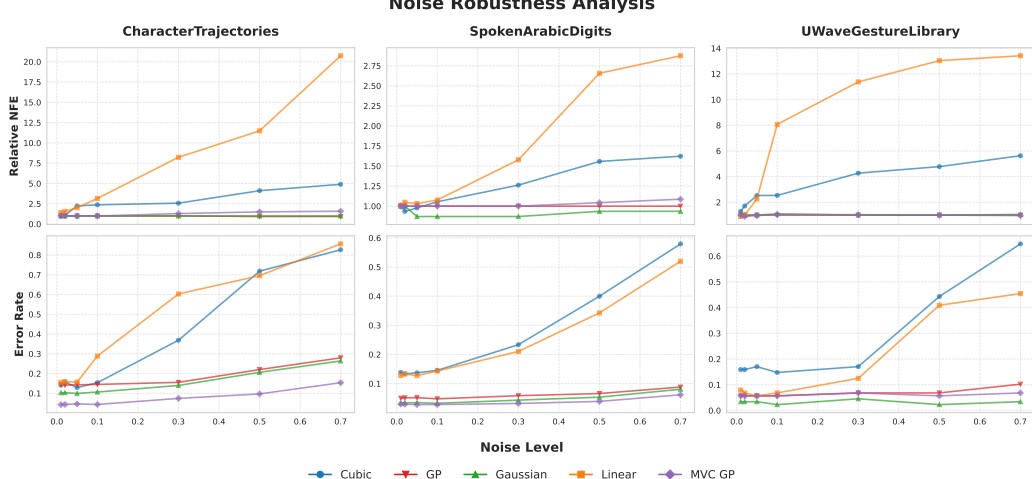

*Figure 4.* **Noise Robustness Analysis.** The plots compare the stability of different path-construction methods under varying intensities of additive white Gaussian noise. *Top row*: Relative NFE normalized to the noise-free number. *Bottom row*: Classification error rate.

longer-range **UWaveGestureLibrary** involves complex human movement patterns. This selection enables us to rigorously assess the robustness of our path-smoothing and attention mechanisms across varying regimes of trajectory roughness, feature dimensionality, and temporal resolution. See Table 2 for detailed numerical results. The proposed **MVC-CDE (GP)** consistently achieves SOTA accuracy and significantly outperforms the baselines. The best models comparison in Figure 3 also supports the evidence that our method attains the best accuracies and efficiency.

### 4.2. Time-Series Forecasting under Missingness

While our primary focus is sequence classification, we also evaluate our method's adaptability to generative forecasting tasks under irregular sampling. We tested models on the ETTm1 dataset with a 24-step forecasting horizon, in which 30% of daily observations were randomly dropped to simulate extreme missingness. To adapt our method for forecasting, we applied it as the encoder, followed by a one-step linear layer, similar to DLinear

As detailed in Table 1, MVC GP outperforms standard Cubic Spline NCDEs in both Mean Squared Error (MSE) and training speed. Furthermore, it yields substantially better predictions than standard GRU baselines, which struggle with the

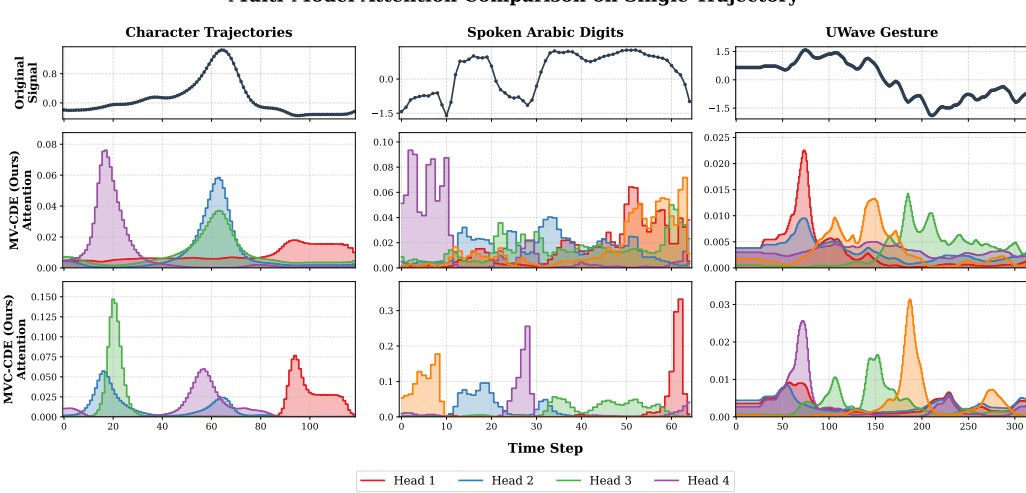

*Figure 5.* **Learned Attention Weights across Datasets. The trajectories visualize the example of learned attention weights for one sample assigned to different smoothing heads over time.**

*Table 1.* **Time-Series Forecasting under Extreme Missingness (ETTm1).** Results for a 24-step horizon with 30% random daily observation drops.

| Model | MSE ↓ | Training Time (s) ↓ |
|---|---|---|
| GRU | $0.476 \pm 0.007$ | $\mathbf{33 \pm 3}$ |
| Neural CDE (Cubic) | $0.487 \pm 0.004$ | $625 \pm 18$ |
| **MVC GP (Ours)** | $\mathbf{0.428 \pm 0.009}$ | $248 \pm 25$ |

continuous-time nature of the missing data. This confirms that our path smoothing and attention mechanism provide robust representational capacity not just for sequence summarization, but also for generative forecasting tasks. Full architectural details and hyperparameters are provided in Appendix B.

Crucially, this forecasting setup also validates the applicability of our method in strictly online, streaming scenarios. While global GP and Kernel smoothing are formally offline operations that require access to the entire trajectory, processing data in rolling causal windows (e.g., fitting the interpolant anew for each history window) enables robust causal forecasting without incurring prohibitive latency. As demonstrated in our computational breakdown (Appendix C.3), the trajectory-fitting overhead remains small, ensuring that the model can be effectively deployed in low-latency environments.

### 4.3. Computational Efficiency

Our method yields substantial training speedups over baselines. As shown in our experiments in Figure 3, MVC-GP demonstrates a speedup ranging from $4.3\times$ to $14.5\times$ compared to standard CDE methods, and up to $16\times$ faster than ODE-RNN on complex tasks like *UWaveGestureLibrary*, while requiring significantly fewer NFE to solve the underlying differential equations. To ensure a fair comparison, we include multiple points for Linear and Cubic baselines by varying their solver tolerances; this demonstrates that their efficiency cannot be matched by simply reducing numerical precision without a significant loss in accuracy.

These results confirm that MVC-GP not only improves classification performance but also ensures high computational efficiency, making it suitable for scalable time-series modeling.

### 4.4. Noise Robustness Analysis

To evaluate the stability of path-construction mechanisms under perturbation, we subject the best-performing models to a robustness test. We inject additive white Gaussian noise into the test set features, sweeping the noise level across a range of intensities. We monitor both the degradation in test

accuracy and the variation in the Average NFE in Figure 4.

The results confirm the hypothesis that smoothing-based methods maintain efficient solver dynamics. As illustrated in Figure 4, direct interpolation methods exhibit a drastic increase in NFE and error rates as noise increases. In contrast, our smoothing approach ensures that the NFE remains almost constant and the error rate degradation is significantly slower, demonstrating superior robustness to noisy inputs.

### 4.5. Impact of Head Number and Smoothing

To understand the scalability of the Multi-View architecture, we analyzed how model performance depends on the number of attention heads and the smoothing bandwidth $h$. In this experiment, all heads are initialized with the same bandwidth $h$, and we observe the effect of increasing $M$ from 1 to 8.

As illustrated in Figure 6, increasing the number of heads generally reduces error rates across all datasets. This effect is most pronounced in configurations with high smoothing parameters, where a single head's focus results in significant information loss due to over-smoothing. By increasing $M$, the attention mechanism enables the solver to leverage multiple representations of the smoothed trajectory, effectively recovering performance.

However, the performance gain follows a law of diminishing returns. We observe that for configurations with GP interpolation, the error rate tends to plateau around $M = 4$.

### 4.6. Interpretability and Attention Dynamics

Figure 5 compares the attention patterns of the trained MV model against the MVC architecture.

**Dataset-Specific Dynamics.** The behavior of the attention heads varies distinctly across datasets, reflecting the nature of the underlying signals:

- **CharacterTrajectories (Low-Dimensional):** The contrast is most visible here: MV attention is blurry, whereas MVC exhibits sharp, localized activations, effectively filtering the trajectory into distinct temporal events.

- **SpokenArabicDigits (High-Dimensional):** As shown in Figure 5, the MV maps are dominated by high-frequency vertical striations induced by the high dimensionality of the input. This pattern indicates that the model reacts to noise and raw signal jumps at individual timestamps. In contrast, MVC smoothes out local noise via convolutions, allowing the attention mechanism to focus on more robust, longer-term features.

- **UWaveGestureLibrary (Long Sequences):** Both

models capture the periodic nature of the gesture data characteristic of long sequences. While the sheer length of the signal makes specific head interpretation less distinct than in shorter tasks, MVC specifically benefits from the input's low dimensionality. This allows it to effectively leverage multi-head capabilities to isolate different aspects of the signal despite the temporal complexity.

### 4.7. Extended Analysis and Ablation Studies

Due to space constraints, comprehensive ablation studies—including evaluations on high-dimensional scalability (963-dimensional PEMS-SF), robustness to high irregular sampling rates (Speech Commands), impact of different adaptive solver orders, head diversity regularization, and comparisons with traditional smoothing splines—are provided in Appendix C.

## 5. Limitations

While our approach significantly improves the efficiency of Neural CDEs, it introduces certain limitations:

- **Solver Dependency:** Our NFE reductions rely explicitly on the step-size control mechanisms of adaptive ODE solvers. The method is not applicable to fixed-step solvers, which cannot dynamically benefit from path smoothness.

- **Bandwidth Sensitivity and Tuning:** Unlike parameter-free exact interpolations, our method requires selecting the bandwidth $h$. While we provide

a log-spaced initialization strategy, finding the optimal bias-variance-NFE trade-off requires validation tuning.

- **Stiffest Head Bottleneck:** In the Multi-View architecture, the global integration step is constrained by the head with the smallest bandwidth.

- **Loss of Frequency Information:** Although our Multi-View architecture mitigates the smoothing-induced information loss, high-frequency features are still inevitably discarded. In cases where these features are not noise, such as long-horizon forecasting, this can lead to significant performance degradation.

## 6. Conclusion

In this work, we addressed the fundamental trade-off between trajectory regularity and representation power in Neural CDEs. We identified that standard interpolation schemes induce numerical stiffness, forcing adaptive solvers to perform excessive function evaluations. To overcome this, we proposed MVC-CDE, a novel architecture that replaces deterministic splines with attentive multi-scale smoothing. By dynamically aggregating information from multiple regularized GP paths, our method effectively decouples the integration cost from input noise. Empirical evaluations on real-world benchmarks demonstrate that MVC-CDE achieves SOTA accuracy while reducing the Number of Function Evaluations by up to an order of magnitude. These results establish path smoothing not merely as a preprocessing heuristic but as a critical architectural component for scalable, efficient continuous-time sequence modeling.

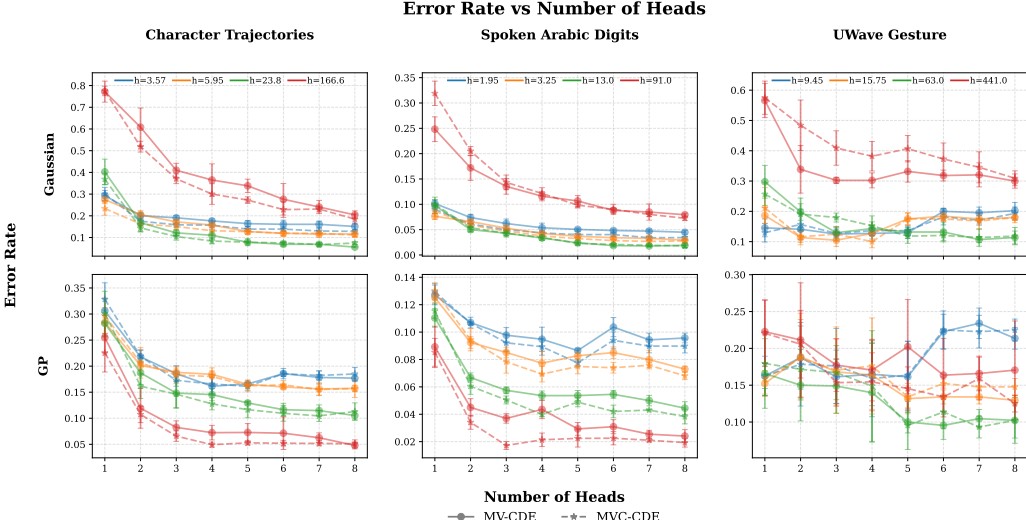

*Figure 6.* **Ablation: Impact of Head Count.** The plots display the error rate as a function of the number of heads across different fixed smoothing bandwidths $h$ for MV-CDE and MVC-CDE.

## Acknowledgements

The work was supported by the grant for research centers in the field of AI provided by the Ministry of Economic Development of the Russian Federation in accordance with the agreement 000000C313925P4F0002 and the agreement №139-10-2025-033.

## Impact Statement

This paper presents work whose goal is to advance the field of Machine Learning. There are many potential societal consequences of our work, none which we feel must be specifically highlighted here.

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

# A. Theoretical Analysis of Path Regularity and Solver Complexity

In this section, we provide a rigorous analysis of the computational complexity of Neural CDEs. We bridge the gap between the analytic properties of the control path $\mathbf{X}(t)$, specifically its Sobolev regularity, and the discrete behavior of adaptive ODE solvers. We derive explicit complexity bounds for deterministic interpolants versus the proposed smoothing priors and formally prove the bottleneck effect in the MV-CDE architecture.

## A.1. ODE Solver Dynamics and Step Size Control

The computational cost of a Neural CDE is dominated by the numerical integration of the latent state $\mathbf{z}(t)$ governed by Eq. (2). Modern implementations employ embedded Runge-Kutta methods, such as the Dormand-Prince pair (Dopri5) of order $p = 5$ (Dormand & Prince, 1980). These solvers dynamically adapt the step size $\eta_i = t_{i+1} - t_i$ to ensure the Local Truncation Error (LTE) remains below a specified tolerance $\delta$.

*Theorem* (3.1). [Step Size dependence on Control Regularity] For a numerical method of order $p$, the asymptotic local truncation error at step $i$ is governed by the principal error function involving the $(p+1)$-th derivative of the solution (Hairer et al., 1993):

$$\text{LTE}_i = C \cdot \eta_i^{p+1} \left\| \frac{\mathrm{d}^{p+1}\mathbf{z}(t)}{\mathrm{d}t^{p+1}} \right\| + \mathcal{O}(\eta_i^{p+2}), \tag{6}$$

where $C$ is a method-dependent constant. Since $\mathbf{z}(t)$ is driven by $\mathbf{X}(t)$, the chain rule implies that higher-order derivatives of the state scale linearly with the higher-order derivatives of the control path. Specifically, assuming the vector field $f_\theta$ is Lipschitz continuous and bounded, we have $\left\| \frac{\mathrm{d}^{p+1}\mathbf{z}}{\mathrm{d}t^{p+1}} \right\| \propto \left\| \frac{\mathrm{d}^{p+1}\mathbf{X}}{\mathrm{d}t^{p+1}} \right\|$. To satisfy the condition $\text{LTE}_i \leq \delta$, the adaptive step size $\eta_i$ must satisfy:

$$\eta_i \leq \left( \frac{\delta}{C \cdot \|\mathbf{X}^{(p+1)}(t_i)\|} \right)^{\frac{1}{p+1}}. \tag{7}$$

Recall that NFE encompasses the total number of calls to the underlying vector field. The total NFE is proportional to the total number of steps required to traverse $[0, T]$. In the continuous limit, incorporating the tolerance constraint, this relates to the integral of the inverse step size up to a constant:

$$\text{NFE} \lesssim \int_0^T \frac{1}{\eta(t)} \mathrm{d}t \lesssim \int_0^T \left( \frac{1}{\delta} \left\| \frac{\mathrm{d}^{p+1}\mathbf{X}(t)}{\mathrm{d}t^{p+1}} \right\| \right)^{\frac{1}{p+1}} \mathrm{d}t. \tag{8}$$

This establishes that NFE is strictly determined by the $L_{1/(p+1)}$-quasi-norm of the $(p+1)$-th derivative of the control path. Unbounded or large derivatives force $\eta \to 0$, causing NFE to diverge.

## A.2. Regularity of Deterministic Interpolation

Standard Neural CDEs utilize linear or natural cubic splines (de Boor, 1978; Reinsch, 1967) to construct $\mathbf{X}(t)$ from discrete observations $\mathcal{S} = \{(t_k, \mathbf{x}_k)\}_{k=1}^N$. We analyze the stiffness of these paths in the presence of input noise.

**Linear Interpolation.** The path $\mathbf{X}_{\text{lin}}(t)$ is Lipschitz continuous ($C^0$) but its derivative is piecewise constant with discontinuities at the knots $t_k$. Adaptive solvers must restart the integration at each discontinuity (Morrill et al., 2022). Moreover, the step size within an interval is constrained by the slope's magnitude. The total integration cost scales with the Total Variation of the sequence:

$$\text{NFE}_{\text{lin}} \lesssim \sum_{k=1}^{N-1} \|\mathbf{x}_{k+1} - \mathbf{x}_k\|. \tag{9}$$

Crucially, for a fixed sampling interval $\Delta t$, if the signal contains additive noise $\epsilon \sim \mathcal{N}(0, \sigma^2)$, the term $\|\mathbf{x}_{k+1} - \mathbf{x}_k\|$ does not vanish but is dominated by the noise amplitude. Thus, $\text{NFE}_{\text{lin}}$ remains high regardless of the underlying signal smoothness.

**Natural Cubic Splines.** The path $\mathbf{X}_{\text{cub}}(t)$ is $C^2$ continuous. However, the interpolation condition $\mathbf{X}(t_k) = \mathbf{x}_k$ forces the spline to traverse the exact geometry of the noise (Wahba, 1990). The stiffness is dominated by the second derivative, which approximates the discrete second-order finite difference:

$$\left\| \frac{\mathrm{d}^2\mathbf{X}_{\text{cub}}(t)}{\mathrm{d}t^2} \right\| \approx \frac{\|\mathbf{x}_{k+1} - 2\mathbf{x}_k + \mathbf{x}_{k-1}\|}{(\Delta t)^2}. \tag{10}$$

Substituting this into the complexity bound yields:

$$\text{NFE}_{\text{cub}} \lesssim \frac{1}{\Delta t} \sum_{k=2}^{N-1} \|\mathbf{x}_{k+1} - 2\mathbf{x}_k + \mathbf{x}_{k-1}\|. \tag{11}$$

This result indicates that cubic splines are extremely sensitive to high-frequency noise, as the cost scales with the local variance normalized by $\Delta t$.

## A.3. Regularity of Smoothing Priors

We contrast the deterministic approach with our proposed Kernel and GP smoothing methods. Let $\mathbf{X}_h(t)$ denote the trajectory obtained via convolution with a smoothing kernel $K_h(t) = \frac{1}{h} K(\frac{t}{h})$ (Wand & Jones, 1995; Tsybakov, 2008).

*Theorem* (3.2). [Derivative Bounds for Smoothing Kernels] Let $\mathbf{X}_h(t)$ be constructed via Nadaraya-Watson regression (Cai, 2001) or as the posterior mean of a Gaussian Process with a stationary RBF kernel with lengthscale $h$ (Rasmussen & Williams, 2006). The $p$-th derivative of the trajectory satisfies the following uniform bound:

$$\sup_{t \in [0,T]} \left\| \frac{\mathrm{d}^p \mathbf{X}_h(t)}{\mathrm{d}t^p} \right\| \le \frac{C_p}{h^p} \|\mathbf{X}\|_\infty, \tag{12}$$

where $C_p = \int |K^{(p)}(u)| du$ is a constant depending solely on the kernel shape, and $\|\mathbf{X}\|_\infty$ is the bound on the input observations.

*Proof.* Both Kernel Smoothing and the GP posterior mean with RBF kernel can be expressed as linear operations on the kernel functions $k_h(t, \cdot)$. Since differentiation is a linear operator and commutes with the integral, we have $\frac{\mathrm{d}^p}{\mathrm{d}t^p} K_h(t) = h^{-p} K^{(p)}(t/h)$ (Wand & Jones, 1995). The bound follows immediately from the properties of the Gaussian kernel, where all derivatives are bounded and scale inversely with the bandwidth $h$. □

*Corollary* (3.3). [Smoothing NFE Complexity] Substituting the bound from Theorem 3.2 into the NFE integral (Theorem 3.1):

$$\text{NFE}(\mathbf{X}_h) \lesssim \int_0^T \delta^{-\frac{1}{p+1}} \left( h^{-(p+1)} \right)^{\frac{1}{p+1}} \mathrm{d}t \lesssim \delta^{-\frac{1}{p+1}} h^{-1}. \tag{13}$$

This confirms that the solver complexity is controlled directly by the hyperparameter $h$. Unlike deterministic splines, the NFE for smoothing priors does not depend on the input noise variance or the sampling grid $\Delta t$, but only on the chosen spectral filter width $h$.

**Bandwidth Selection and Bias-Variance-NFE Trade-off.** The selection of the bandwidth $h$ is critical. Classically, optimal bandwidths (e.g., $h \propto n^{-1/5}$ according to Silverman's rule (Silverman, 2018)) are chosen to minimize the Asymptotic Mean Integrated Squared Error (AMISE). However, in the context of Neural CDEs, these bandwidths often retain high-frequency local fluctuations that are artifacts of input noise rather than signal.

For an adaptive ODE solver, these fluctuations act as high-frequency drivers that force the solver to take excessively small steps ($\eta_i \to 0$), leading to a massive increase in NFE. Consequently, classical statistical optimality trades off poorly against computational budget. Our approach treats $h$ as a task-aware hyperparameter tuned on a validation set. This allows for a principled increase in bandwidth ($h > c \cdot n^{-1/5}$), which deliberately oversmooths the path to dramatically lower NFEs.

In our Multi-View architecture, this oversmoothing is mitigated by the ensemble: while individual heads may be overly smooth, the learnable query mechanism allows the model to selectively attend to the original raw inputs or aggregate multi-scale features, effectively recovering the task-relevant details that might be obscured by macroscopic smoothing in a single-head model.

## A.4. Complexity of Parallel Multi-Head Integration

We analyze the Multi-View architecture (MV-CDE and MVC-CDE), where the global state $\mathbf{Z}(t) = [\mathbf{z}^{(1)}(t), \ldots, \mathbf{z}^{(M)}(t)]^\top$ involves $M$ parallel latent trajectories. Each head $m$ is driven by a control path $\mathbf{X}^{(m)}$ with a specific lengthscale $h_m$.

*Theorem* (3.4). [Parallel Integration Bottleneck] Consider an adaptive ODE solver integrating the block-diagonal system $\mathbf{Z}(t)$ with a global error tolerance $\delta$. The total NFE scales jointly with the solver tolerance and the minimum smoothing parameter in the ensemble as:

$$\text{NFE}_{\text{total}} \lesssim \delta^{-\frac{1}{p+1}} \left( \min_{m=1}^{M} h_m \right)^{-1}. \tag{14}$$

*Proof.* Standard adaptive solvers (e.g., `dopri5` in `torchdiffeq`) control the error using a unified norm on the concatenated state vector. Let $\mathbf{e}_i \in \mathbb{R}^{\sum d_m}$ be the estimated local error vector at step $i$, composed of sub-vectors $\mathbf{e}_i^{(m)}$ for each head (Hairer et al., 1993). The step acceptance criterion is $\|\mathbf{e}_i\| \leq \delta$. Using the infinity norm (or broadly any $p$-norm for finite dimensions), this condition implies that the error in *every* component must satisfy the tolerance:

$$\forall m \in \{1, \dots, M\}: \quad \|\mathbf{e}_i^{(m)}\| \lesssim \delta. \tag{15}$$

From Theorem 3.1 and Theorem 3.2, the error estimate for the $m$-th head scales as:

$$\|\mathbf{e}_i^{(m)}\| \approx C \cdot \eta_i^{p+1} \cdot h_m^{-(p+1)}. \tag{16}$$

To ensure $\|\mathbf{e}_i^{(m)}\| \leq \delta$ for all $m$ simultaneously, the global step size $\eta_i$ must satisfy the tightest constraint:

$$\eta_i \leq \min_{m=1}^{M} \left( \frac{\delta}{C} \cdot h_m^{p+1} \right)^{\frac{1}{p+1}} = C^{-\frac{1}{p+1}} \delta^{\frac{1}{p+1}} \min_{m=1}^{M} h_m \lesssim \delta^{\frac{1}{p+1}} \min_{m=1}^{M} h_m. \tag{17}$$

The global step size is therefore limited by the trajectory with the highest stiffness, which corresponds to the smallest lengthscale $h_{\min} = \min_m h_m$. Integrating over the interval $[0, T]$ yields the joint scaling law for the total computational cost:

$$\text{NFE}_{\text{total}} \lesssim \int_0^T \frac{1}{\eta(t)} \mathrm{d}t \lesssim \delta^{-\frac{1}{p+1}} (h_{\min})^{-1}. \tag{18}$$

Thus, the computational efficiency of the MV-CDE and MVC-CDE models is bottlenecked by the finest temporal scale represented in the multi-view ensemble. $\square$

## A.5. Explicit Error Comparison

To strictly quantify the performance gap, we evaluate the Mean Squared Error (MSE) of the estimator $\hat{\mathbf{X}}(t)$ with respect to the true latent process $\mathbf{X}(t)$. Since both Gaussian Process regression and Nadaraya-Watson (Kernel) smoothing are linear smoothers, they can be written as $\hat{\mathbf{X}}(t) = \mathbf{w}(t)^\top \mathbf{x}$, where $\mathbf{w}(t)$ is a weight vector.

**Gaussian Process Error.** The GP weights $\mathbf{w}_{\text{GP}} = (\mathbf{K} + \sigma^2 \mathbf{I})^{-1} \mathbf{k}(t)$ are derived analytically by solving the optimization problem $\min_{\mathbf{w}} \mathbb{E}[\|\mathbf{X}(t) - \mathbf{w}^\top \mathbf{x}\|^2]$. The resulting minimum error variance is:

$$\mathcal{E}_{\text{GP}}(t) = k(t, t) - \mathbf{k}(t)^\top (\mathbf{K} + \sigma^2 \mathbf{I})^{-1} \mathbf{k}(t). \tag{19}$$

This expression represents the theoretical lower bound on the reconstruction error for any linear estimator under the assumed prior (Rasmussen & Williams, 2006).

**Kernel Smoothing Error.** The Nadaraya-Watson weights are heuristic: $\mathbf{w}_{\text{NW}} = \mathbf{k}(t)/(\mathbf{k}(t)^\top \mathbf{1})$. The error for this sub-optimal weight selection is strictly larger:

$$\mathcal{E}_{\text{NW}}(t) = \mathcal{E}_{\text{GP}}(t) + \underbrace{(\mathbf{w}_{\text{NW}} - \mathbf{w}_{\text{GP}})^\top (\mathbf{K} + \sigma^2 \mathbf{I})(\mathbf{w}_{\text{NW}} - \mathbf{w}_{\text{GP}})}_{\text{Excess Error} > 0}. \tag{20}$$

The Excess Error term is a quadratic form with a positive-definite matrix, meaning $\mathcal{E}_{\text{NW}}(t) > \mathcal{E}_{\text{GP}}(t)$ everywhere, except in trivial cases. This mathematically confirms that for a fixed smoothing bandwidth $h$, the GP provides a strictly more accurate reconstruction of the control path.

# B. Experiments Setup

In this section, we detail the experimental setup used to evaluate the proposed MV-CDE and MVC-CDE architectures. Our evaluation focuses on the robustness of path-construction methods, their impact on the numerical efficiency of the ODE solver, and the resulting classification performance compared to baselines.

### B.1. Datasets

We selected three multivariate time series classification benchmarks from the UEA (Bagnall et al., 2018) and UCR (Dau et al., 2019) repositories. These datasets exhibit diverse characteristics in terms of sequence length, dimensionality, and signal regularity.

**CharacterTrajectories.** This dataset consists of 2,858 sequences recording the trajectory of a pen tip during handwriting (3 features: $x, y$-coordinates, pen tip force). The maximum sequence length is 182 time steps. The task is to classify inputs into 20 distinct characters.

**SpokenArabicDigits.** This dataset comprises 8,800 time series representing MFCCs extracted from spoken Arabic digits. It features high dimensionality (13 features) and highly variable sequence lengths (ranging from 4 to 93 frames). The task is a 10-class digit classification.

**UWaveGestureLibrary.** This dataset contains 4,480 sequences of accelerometer gesture data (3 continuous spatial dimensions: $x, y, z$). The sequence length is fixed at 315 time steps, and the objective is to distinguish between 8 gesture patterns.

To evaluate our model's robustness and scalability in the supplementary experiments, we utilized three additional datasets with the following configurations:

**ETTm1 (Electricity Transformer Temperature):** This dataset consists of 69,680 continuous measurements of electricity transformer metrics across 7 features (oil temperature and 6 power load indicators). For our generative forecasting task under extreme missingness (simulated by randomly dropping 30% of observations), the data is windowed into multivariate sequences of length 24 to predict a 24-step future horizon.

**Speech Commands:** This dataset comprises 34,975 one-second audio recordings of spoken words processed into continuous sequences (20 MFCC features). The sequence length is fixed at 161 time steps, and the objective is to classify the input into one of 10 spoken word categories. We utilized this setup to evaluate model robustness under high irregular sampling rates (with 30% and 50% random drop rates).

**PEMS-SF:** This dataset contains 440 daily freeway occupancy rate records in the San Francisco Bay Area across 963 spatial sensor channels. The sequence length is fixed at 144 time steps (representing 10-minute intervals over 24 hours), and the task is to classify each day into 7 distinct days of the week. It is used specifically to benchmark the memory and computational scalability of continuous-time models in extremely high spatial dimensions.

### B.2. Data Preprocessing

To ensure rigorous evaluation and reproducibility, we employed a standardized preprocessing pipeline across all experiments. We partitioned each dataset into training (60%), validation (20%), and testing (20%) sets. All reported results are averaged over five independent random seeds.

Channel-wise Z-score normalization was applied using statistics computed solely on the training split to prevent data leakage. To accommodate variable-length sequences within batched continuous-time operations, we padded inputs to the maximum sequence length using the final observation value (rectilinear padding). This strategy yields a zero derivative ($dX/dt = 0$) in padded regions, allowing the adaptive solver to traverse them efficiently. The input features remain purely spatial ($x_t \in \mathbb{R}^d$), and the time domain is defined as regular intervals $t \in \{0, 1, \ldots, T\}$ with $\Delta t = 1$.

### B.3. Implementation Details and General Setup

All models were implemented in PyTorch and trained on a single NVIDIA L40 GPU using the Adam optimizer to minimize Cross-Entropy loss. The best models were selected based on validation accuracy checkpointing.

To ensure a fair computational comparison, the vector field $f_\theta$ for all ODE-based variants was parameterized by a Multilayer Perceptron (MLP) with `tanh` activations. Crucially, rather than using a fixed hidden dimension across all models, we meticulously adjusted the hidden layer dimensions of the Linear and Cubic CDE baselines to exactly match the total parameter count of our proposed multi-view architectures. This ensures that any performance gains are attributed to the architectural design and path construction, rather than an arbitrary difference in model capacity. We used the `dopri5` adaptive solver.

To quantify performance and efficiency, we utilize the following metrics:

- **Total wall-clock time** is calculated as the sum of: (i) trajectory fit time, (ii) training duration, and (iii) the final test evaluation pass.

- **Average NFE per batch** is reported during the test phase to assess solver efficiency. This metric reflects the numerical complexity of integrating the stiffest trajectory within a batch.

### B.4. Baselines and Hyperparameter Optimization

**Non-CDE Baselines (ODE-RNN, GRU-D).** We compare our methods against standard recurrent architectures capable of modeling continuous dynamics.

- **ODE-RNN (Rubanova et al., 2019):** The output layer of the ODE function is initialized with zeros ($dh/dt \approx 0$ initially) for stability.

- **GRU-D (Che et al., 2016):** The decay layer weights are initialized to zero ($\gamma = 1$ initially) to prevent signal vanishing at the start of training.

- **Mamba (Gu & Dao, 2024):** A recent linear-time state-space model that bypasses ODE solvers entirely. We configured it with standard parameters (e.g., $d_{\text{state}} = 16$, $n_{\text{layers}} = 2$, utilizing parallel scans).

Unlike our proposed methods, these baselines required distinct hyperparameter tuning to achieve convergence. We optimized the learning rate (e.g., searching within $\{10^{-3}, 5 \cdot 10^{-3}, 10^{-2}\}$) and solver tolerance ($\{10^{-2}, 10^{-3}\}$) specifically for each dataset. For instance, ODE-RNN required a learning rate of 0.01 on SpokenArabicDigits and 0.005 on CharacterTrajectories to match the best performance.

**Log-Neural CDE.** This baseline (Walker et al., 2024) interpolates Log-Signatures computed over fixed time windows. Similar to the non-CDE baselines, we found that this model required higher learning rates than standard NCDEs. We optimized the window step size from $\{3, 5, 10, 20, 30\}$, Log-Signature depth $\{1, 2\}$, and the learning rate ($\{5 \cdot 10^{-3}, 10^{-2}\}$).

**Standard Neural CDEs (Linear & Cubic)** (Kidger et al., 2020) These models use fixed paths constructed via Linear Interpolation or Natural Cubic Splines. Crucially, to strictly evaluate the impact of path construction, these models utilized the exact same optimization schedule and learning rates as our proposed methods (detailed in Section B.6).

### B.5. Proposed Methods

**Kernel CDE & GP CDE.** These models employ smooth path construction using Nadaraya-Watson kernel smoothing or Gaussian Process posterior means to generate a single continuous trajectory per sample. Unlike the multi-view methods, these do not utilize attention mechanisms. We evaluated GP observation noise across $\sigma \in \{0.01, 0.05\}$.

**Smoothing Splines:** To contrast our attentive multi-view approach with traditional global smoothing, we implemented a baseline using SciPy's `UnivariateSpline`, which penalizes the second derivative based on a fixed global smoothing factor before computing CDE coefficients.

**MV-CDE & MVC-CDE.** These are the core contributions of this work, where $M \geq 1$ trajectories are constructed dynamically.

- **MV-CDE:** Uses $M$ learnable query vectors $q_m \sim \mathcal{N}(0, 1)$ to compute attention weights and construct $M$ smooth trajectories via weighted interpolation.

- **MVC-CDE:** Incorporates a feature extraction stage (2-layer 1D CNN, kernel size 3, 128 hidden units) before attention computation. The attention weights are derived from latent features, while the continuous path interpolates the original input space.

**Global Smoothing Strategy.** For all smoothing-based models, we swept over raw bandwidth factors in the range $[0.01, 9.0]$. The effective smoothing scale is adapted to the sequence length, such that $h_{\text{eff}} = h_{\text{raw}} \times T_{\text{max}}$.

## B.6. Dataset-Specific Optimization Settings

The following settings were applied uniformly to Linear, Cubic, and our proposed models. As noted in Section B.4, other baselines relied on separately optimized parameters.

- **CharacterTrajectories:** Batch size 256, learning rate $10^{-4}$, weight decay $10^{-5}$, 25 epochs. Solver tolerance $10^{-3}$.

- **SpokenArabicDigits:** Batch size 2048, learning rate $10^{-3}$, weight decay $10^{-4}$, 13 epochs. Solver tolerance $10^{-3}$.

- **UWaveGestureLibrary:** Batch size 512, learning rate $10^{-3}$, weight decay $10^{-5}$, 30 epochs. Solver tolerance $10^{-4}$.

- **ETTm1 (Forecasting):** Context length 24, forecasting horizon 24. Learning rate $10^{-3}$ (AdamW optimizer), MSE loss, solver tolerance $10^{-4}$, 4 attention heads for ConvCDE. To strictly isolate the encoder's representational power, we avoided complex autoregressive decoders. Instead, we employed Reversible Instance Normalization (RevIN)(Kim et al., 2021) to mitigate distribution shifts. The encoder (MVC, Neural CDE, or GRU) maps the input to an embedding, which is directly projected to the 24-step horizon via a simple linear head before RevIN denormalization.

- **Speech Commands:** Batch size 512, learning rate $10^{-3}$, 20 epochs. Solver tolerance $10^{-2}$. A time scaling factor of 10 was applied to stabilize solver dynamics.

- **PEMS-SF:** Batch size 32, learning rate $10^{-3}$, 30 epochs. Solver tolerance $10^{-3}$. (The explicit time channel was omitted due to the highly regular and synchronous nature of the underlying spatial grid).

## B.7. Ablation Studies and Visualization

**Solver Tolerance Analysis.** To investigate speed-accuracy trade-offs specifically for spline-based methods, we conducted ablation studies on the solver tolerance for the Linear and Cubic Neural CDE baselines. We varied the absolute and relative tolerances across $\{10^{-1}, 10^{-2}, 10^{-3}, 10^{-4}\}$ to determine whether relaxing precision constraints could accelerate these models without catastrophic accuracy loss. These results are shown in Figures 7, 8, and 9.

**Impact of Head Count and Smoothing.** We conducted comprehensive ablation studies varying the number of attention heads $M \in [1, 8]$. We tested two smoothing configurations: (1) *homogeneous*, where the bandwidth $h$ is identical across all heads; and (2) *heterogeneous*, where $h$ is distinct for each head (initialized with geometrically spaced values). This comparison, visualized in Figure 6, assesses the solver's ability to integrate systems with multiscale stiffness.

**Noise Robustness Analysis.** To evaluate the stability of path-construction mechanisms under perturbation, we subject the best-performing models to a robustness test. We inject additive white Gaussian noise $\xi \sim \mathcal{N}(0, \lambda^2 I)$ into the test set features, sweeping the noise level $\lambda$ across a range of intensities. We monitor both the degradation in test accuracy and the variation in the Average NFE. This experiment is designed to verify the hypothesis that smoothing-based methods (Kernel, GP) maintain efficient solver dynamics by filtering high-frequency jitter, whereas direct interpolation methods (Linear, Cubic) transmit noise directly into the vector field, forcing the adaptive solver to take excessive steps to track the irregularities.

**Attention Visualization.** We explicitly extracted the attention weights from trained MV-CDE and MVC-CDE models. This enables a qualitative analysis of temporal feature selection, verifying whether the multi-head mechanism effectively focuses on informative trajectory segments while suppressing noise.

**Generative Time-Series Forecasting.** To evaluate our method's capacity for sequence prediction under incomplete data, we set up a forecasting task using the ETTm1 dataset. We simulate random missingness by dropping 30% of the observation points. Models are trained using a historical context length of 24 time steps to predict a future horizon of 24 steps, comparing MVC GP against standard Cubic Splines and discrete GRU baselines. Global GP and kernel smoothing are formally offline operations over each history window; in this experiment we do not precompute interpolants across the dataset—we fit the control path anew for each batch from the 24-step context only, which enforces a strict causal boundary between history and horizon.

We emphasize that this trade-off between offline path construction and computational efficiency is not unique to our method. As discussed by Morrill et al. (2022), most standard Neural CDE implementations already rely on non-causal interpolants (e.g., natural cubic splines fitted with access to the full sequence), and are therefore not strictly suited to step-by-step streaming in the first place. Our contribution targets the orthogonal bottleneck of solver stiffness and NFE. In operational

forecasting, a practical deployment pattern is to apply attentive smoothing over rolling causal windows—accepting batch- or window-level latency rather than single-step updates—which the ETTm1 setup is designed to approximate.

**Attention Head Diversity and Regularization.** To measure the distinctiveness of the learned paths, we quantify diversity among attention heads on the UWaveGestureLibrary dataset. We evaluate the maximum pairwise Cosine Distance and Jensen-Shannon Divergence across the temporal attention maps. Additionally, we set up an optimization test introducing a similarity-penalizing loss term $\mathcal{L}_{\text{reg}}$ with a coefficient $\lambda = 0.1$ to evaluate whether explicitly forcing head diversity improves or hinders representational capacity.

**Robustness to Sparse Grids (Irregular Sampling).** To test CDE performance under high rates of missing data, we subject our architectures and standard spline baselines to a drop rate sweep of 30% and 50% on the Speech Commands dataset. We monitor the stability of the Average NFE and classification accuracy as the sampling density decreases.

**Adaptive ODE Solver Order.** To assess the numerical sensitivity of our smoothed paths compared to deterministic splines, we evaluate model dynamics on the UWaveGestureLibrary dataset using different embedded Runge-Kutta adaptive solvers: `bosh3` (3rd order), `dopri5` (5th order), and `dopri8` (8th order). We track total training time and average NFEs to determine the optimal solver order for regularized trajectories.

**High-Dimensional Computational Scalability.** To evaluate the limits of continuous-time integration, we benchmark our method on the 963-dimensional PEMS-SF dataset. We record training times and average NFEs, comparing our smooth multi-scale path formulation directly against the hardware-accelerated discrete parallel scan of the Mamba state-space architecture.

**Smoothing Spline Baseline Comparison.** To demonstrate that our model performs a task-aware dynamic representation rather than a basic filtering operation, we benchmark our architecture against a traditional Smoothing Spline. Using SciPy's `UnivariateSpline`, we globally smooth the raw sequences with a length-based penalty before applying standard cubic CDE coefficients, and compare the resulting accuracy across CharacterTrajectories, SpokenArabicDigits, and UWaveGestureLibrary.

## C. Full Experimental Results and Memory Complexity

In this section, we provide supplementary experimental analysis to support the claims made in the main text. Specifically, we analyze the limitations of single-kernel baselines, investigate the impact of smoothing initialization strategies on the accuracy-efficiency trade-off, and provide a granular breakdown of computational costs.

Table 2 presents the comprehensive comparison of all evaluated models, including precise hyperparameters used to achieve the reported results.

Additionally, Table 3 details the total parameter counts and peak GPU memory (VRAM) usage for the evaluated models. To ensure a fair comparison, the baseline CDEs were strictly capacity-matched to our MVC GP architecture. While our multi-view approach introduces a moderate memory overhead compared to standard CDEs—due to the simultaneous integration of multiple paths—it remains exceptionally memory-efficient compared to discrete state-space models. Specifically, Mamba's hardware-aware parallel scan, while computationally fast, requires materializing large hidden state expansions across the entire sequence length. This results in peak memory usage up to $50\times$ higher than that of our continuous-time approach for long or high-dimensional datasets (e.g., 62 GB vs 1 GB on SpokenArabicDigits). This highlights a significant practical advantage of our smoothed CDE formulation in memory-constrained environments.

**Component-wise decomposition.** Table 2 supports a direct ablation of our design choices on the UEA benchmarks. Replacing splines with single-path smoothing (Gaussian or GP) typically reduces NFE by roughly $3\times$–$5\times$ while maintaining competitive accuracy (e.g., on UWaveGestureLibrary, cubic NFE $\approx 1042$ vs. GP $\approx 171$). Adding the multi-view attention mechanism (MV over single-path GP) yields an additional accuracy gain on the order of $1\%$–$4\%$ (e.g., MV GP 92.95% vs. GP 91.14% on UWave). The convolutional front-end (MVC) contributes a further $\approx 2\%$–$3\%$ in accuracy and can halve total training time when combined with coarse bandwidths (e.g., CharacterTrajectories: MV GP 92.97% / 46.2 s vs. MVC GP 95.07% / 22.6 s). GP smoothing generally attains higher accuracy than kernel smoothing at comparable NFE, but kernel fitting is cheaper; GP models can require up to $\approx 2\times$ longer training on some datasets when finer scales are needed.

*Table 2.* Performance comparison of various models across CharacterTrajectories, SpokenArabicDigits, and UWaveGestureLibrary datasets. Our proposed methods are marked with (Ours), with **MVC GP** being our primary architecture. Best results in each category are highlighted in **bold** (for NFE, the best non-zero value is highlighted).

| Model | Test Acc. (%) | Total Time (s) | Avg NFE | Hyperparameters |
|---|---|---|---|---|
| **CharacterTrajectories** | | | | |
| Linear | 90.59 ± 1.47 | 129.02 ± 1.63 | 347.00 ± 13.49 | LR:0.0001 — Tol:0.001 |
| Cubic | 91.89 ± 0.47 | 100.22 ± 0.44 | 239.00 ± 4.69 | LR:0.0001 — Tol:0.001 |
| ODE-RNN | 74.65 ± 4.07 | 333.62 ± 23.88 | 1953.03 ± 317.93 | LR:0.005 — Tol:0.001 |
| GRU-D | 93.17 ± 0.78 | **10.56 ± 0.39** | 0.00 ± 0.00 | LR:0.005 — Tol:0.001 |
| Log-NCDE | 90.87 ± 0.89 | 163.57 ± 1.96 | 839.86 ± 26.51 | LR:0.005 — Tol:0.001 — Depth:1.0 — Step:10.0 |
| Mamba | **96.64 ± 1.24** | 31.16 ± 0.13 | 0.00 ± 0.00 | LR:0.009 — Tol:0.001 — L:2 — DSt:16 — Exp:2 — Pscan:True |
| Gaussian (Ours) | 89.20 ± 2.26 | 49.53 ± 2.24 | 165.63 ± 4.23 | BW:5.95 — LR:0.0001 — Tol:0.001 |
| GP (Ours) | 88.15 ± 0.42 | 46.41 ± 1.61 | 169.23 ± 2.06 | LR:0.0001 — Tol:0.001 — LS:71.4 — Noise:0.01 |
| MV Gaussian (Ours) | 89.93 ± 0.73 | 50.31 ± 2.78 | 190.49 ± 10.77 | BW:[3.57,11.9,47.6,166.6] — LR:0.0001 — Tol:0.001 |
| MV GP (Ours) | 92.97 ± 1.74 | 46.20 ± 1.65 | 206.60 ± 5.60 | BW:[3.57,11.9,47.6,166.6] — LR:0.0001 — Tol:0.001 — Noise:0.01 |
| MVC Gaussian (Ours) | 91.71 ± 1.40 | 51.65 ± 1.63 | 209.34 ± 15.26 | BW:[3.57,11.9,47.6,166.6] — LR:0.0001 — Tol:0.001 — KS:3.0 |
| **MVC GP (Ours, Main)** | 95.07 ± 0.49 | 22.63 ± 0.48 | **95.17 ± 7.41** | BW:[166.6,166.6,166.6,166.6] — LR:0.0001 — Tol:0.001 — Noise:0.01 — KS:3.0 |
| **SpokenArabicDigits** | | | | |
| Linear | 89.44 ± 0.93 | 120.51 ± 1.29 | 1119.00 ± 42.00 | LR:0.001 — Tol:0.001 |
| Cubic | 88.70 ± 1.03 | 42.74 ± 0.84 | 339.00 ± 7.35 | LR:0.001 — Tol:0.001 |
| ODE-RNN | 68.47 ± 2.62 | 53.29 ± 4.81 | 1359.20 ± 170.47 | LR:0.01 — Tol:0.001 |
| GRU-D | 96.57 ± 0.13 | **2.02 ± 0.09** | 0.00 ± 0.00 | LR:0.01 — Tol:0.001 |
| Log-NCDE | 86.81 ± 0.54 | 11.32 ± 0.16 | 215.80 ± 9.23 | LR:0.01 — Tol:0.001 — Depth:1.0 — Step:20.0 |
| Mamba | 96.82 ± 0.94 | 44.82 ± 16.73 | 0.00 ± 0.00 | LR:0.005 — Tol:0.001 — L:2 — DSt:16 — Exp:2 — Pscan:True |
| Gaussian (Ours) | 96.32 ± 0.61 | 14.27 ± 0.58 | 187.00 ± 5.10 | BW:3.25 — LR:0.001 — Tol:0.001 |
| GP (Ours) | 96.12 ± 0.33 | 11.73 ± 0.46 | 181.40 ± 5.37 | LR:0.001 — Tol:0.001 — LS:39.0 — Noise:0.01 |
| MV Gaussian (Ours) | 97.11 ± 0.27 | 19.59 ± 0.68 | 195.40 ± 6.54 | BW:[1.95,6.5,26.0,91.0] — LR:0.001 — Tol:0.001 |
| MV GP (Ours) | 97.52 ± 0.39 | 13.61 ± 0.48 | 187.00 ± 7.21 | BW:[6.5,26.0,91.0] — LR:0.001 — Tol:0.001 — Noise:0.01 |
| MVC Gaussian (Ours) | 97.34 ± 0.46 | 19.12 ± 0.56 | 191.40 ± 9.94 | BW:[1.95,6.5,26.0,91.0] — LR:0.001 — Tol:0.001 — KS:3.0 |
| **MVC GP (Ours, Main)** | **98.28 ± 0.29** | 7.32 ± 0.15 | **95.40 ± 0.89** | BW:[91.0,91.0,91.0] — LR:0.001 — Tol:0.001 — Noise:0.01 — KS:3.0 |
| **UWaveGestureLibrary** | | | | |
| Linear | 86.82 ± 4.22 | 134.34 ± 2.17 | 2068.20 ± 151.75 | LR:0.001 — Tol:0.0001 |
| Cubic | 85.68 ± 3.47 | 81.78 ± 5.36 | 1042.20 ± 116.76 | LR:0.001 — Tol:0.0001 |
| ODE-RNN | 37.95 ± 2.06 | 205.21 ± 1.27 | 4397.20 ± 2.68 | LR:0.005 — Tol:0.01 |
| GRU-D | 78.86 ± 2.62 | **5.74 ± 0.11** | 0.00 ± 0.00 | LR:0.01 — Tol:0.001 |
| Log-NCDE | 70.91 ± 6.36 | 34.87 ± 0.53 | 753.80 ± 25.60 | LR:0.01 — Tol:0.001 — Depth:1.0 — Step:30.0 |
| Mamba | 80.23 ± 8.26 | 17.47 ± 0.08 | 0.00 ± 0.00 | LR:0.005 — Tol:0.001 — L:2 — DSt:16 — Exp:2 — Pscan:True |
| Gaussian (Ours) | 88.41 ± 2.59 | 14.84 ± 0.80 | 236.60 ± 6.84 | BW:15.75 — LR:0.001 — Tol:0.0001 |
| GP (Ours) | 91.14 ± 1.87 | 8.69 ± 0.25 | **170.60 ± 5.37** | LR:0.001 — Tol:0.0001 — LS:441.0 — Noise:0.05 |
| MV Gaussian (Ours) | 89.77 ± 3.31 | 17.97 ± 1.79 | 320.60 ± 23.08 | BW:[9.45,31.5,126.0,441.0] — LR:0.001 — Tol:0.0001 |
| MV GP (Ours) | 92.95 ± 1.24 | 17.19 ± 1.20 | 359.00 ± 23.62 | BW:[15.75,63.0,189.0] — LR:0.001 — Tol:0.0001 — Noise:0.05 |
| MVC Gaussian (Ours) | 90.00 ± 2.03 | 19.87 ± 1.78 | 453.80 ± 38.28 | BW:[15.75,15.75,15.75,15.75] — LR:0.001 — Tol:0.0001 — KS:3.0 |
| **MVC GP (Ours, Main)** | **95.39 ± 0.28** | 12.29 ± 0.27 | 218.60 ± 10.04 | BW:[31.5,63.0,126.0] — LR:0.001 — Tol:0.0001 — Noise:0.05 — KS:3.0 |

*Table 3.* **Comparison of Parameter Counts and Peak Memory.** Values are presented as Parameters / Peak VRAM (MB). Models were capacity-matched by parameter count.

| Model | CharacterTrajectories | SpokenArabicDigits | UWaveGestureLibrary |
|---|---|---|---|
| Linear (NCDE) | 325,704 / 152.37 | 753,686 / 485.73 | 253,096 / 140.07 |
| Cubic (NCDE) | 325,704 / 155.06 | 753,686 / 510.18 | 253,096 / 143.29 |
| MVC Gaussian (Ours) | 326,040 / 160.12 | 986,510 / 1,070.26 | 319,884 / 286.04 |
| **MVC GP (Ours)** | 326,040 / 375.26 | 753,933 / 1,075.16 | 252,683 / 1,710.11 |
| Mamba | 326,820 / 8,478.39 | 754,738 / 62,323.94 | 249,488 / 25,224.74 |

## C.1. Single-Kernel Smoothing Baselines

To verify that the performance gains stem from the Multi-View Attention mechanism rather than from the use of Gaussian Processes alone, we evaluated single-trajectory models without attention.

Figure 7 presents the performance landscape (Error Rate vs. Training Time) for single-kernel methods compared to standard baselines. While single-kernel models often outperform ODE-RNN in speed, they exhibit high accuracy variance across different bandwidth choices. This instability underscores the necessity of the Multi-View mechanism to robustly aggregate dynamics.

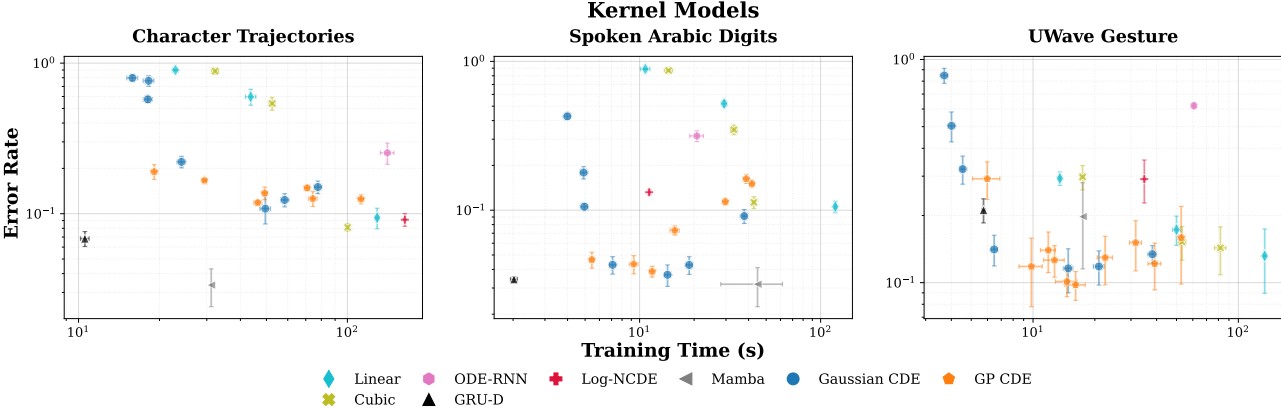

*Figure 7.* **Single-Kernel Smoothing Performance.** The scatter plots illustrate the trade-off between error rate and training time for single-kernel Gaussian and GP models.

## C.2. Impact of Bandwidth Initialization Strategy

We further investigate how the diversity of smoothing scales impacts the model's position on the Pareto frontier. We compare two initialization configurations:

- **Heterogeneous Smoothing (Figure 8):** Bandwidths are initialized with geometrically spaced values to capture multi-scale dynamics.

- **Homogeneous Smoothing (Figure 9):** All heads share the same initial bandwidth.

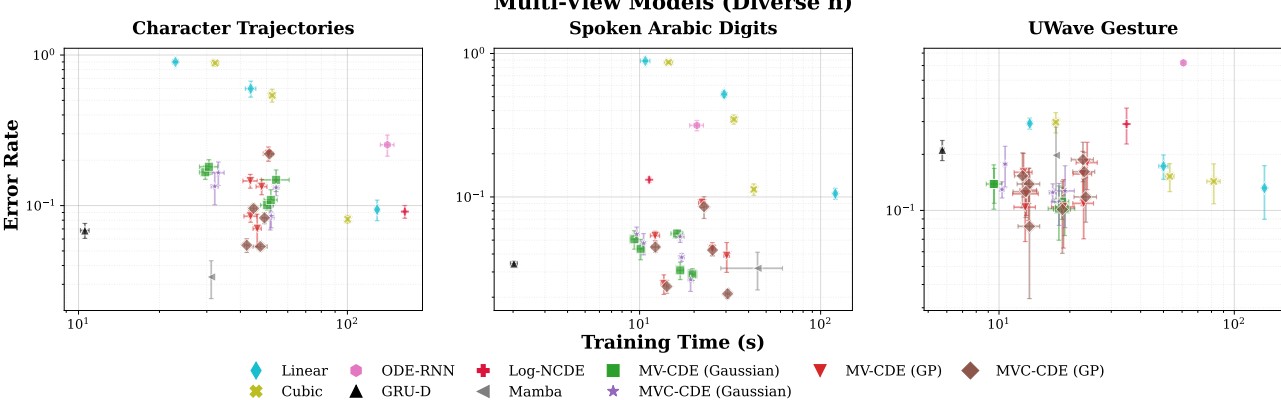

*Figure 8.* **Pareto Efficiency: Heterogeneous Bandwidths.** Error Rate vs. Training Time comparison when attention heads are initialized with diverse smoothing scales.

**Analysis.** As shown in Figure 8, the Heterogeneous strategy pushes the proposed models (MVC-GP and MVC-Gaussian) towards the bottom-left corner of the plot, indicating superior accuracy with minimal training time. This diversity enables

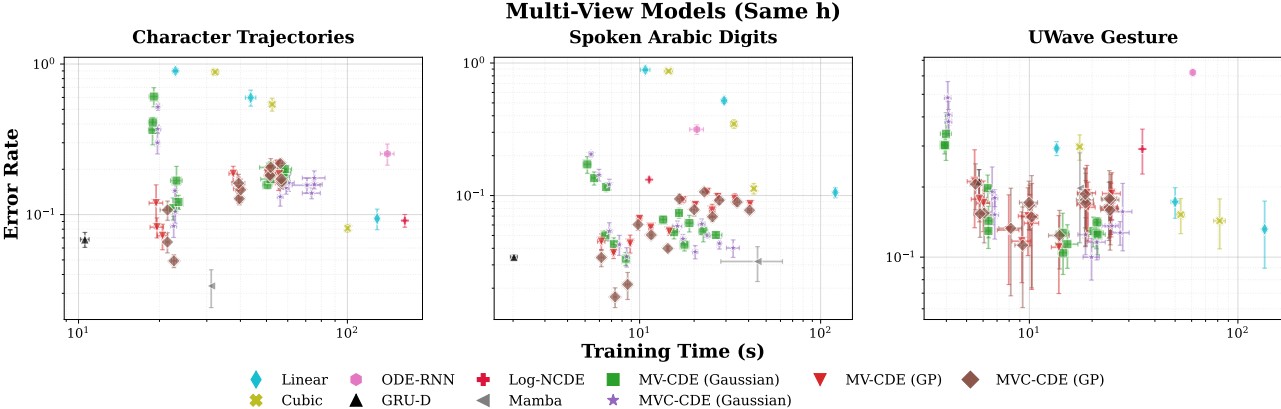

*Figure 9.* **Pareto Efficiency: Homogeneous Bandwidths.** Performance landscape when all attention heads are initialized with identical smoothing scales.

the model to effectively disentangle stiff and smooth dynamics. In contrast, while the Homogeneous setting (Figure 9) still outperforms standard Linear and Cubic baselines, the lack of scale diversity yields a less favorable trade-off, with models often clustering at higher error rates or requiring longer training times than their heterogeneous counterparts.

### C.3. Detailed Computational Cost Analysis

To explain the speedups reported in the main text, Figure 10 provides a granular breakdown of wall-clock time measured entirely on an NVIDIA H100 with small batch sizes, which better match online and streaming workloads. We profile on the H100 because per-trajectory GP fitting is embarrassingly parallel and small batches avoid saturating the device's parallelization capacity. We distinguish between:

1. **Pure Training Time:** The time spent by the ODE solver during the iterative training loop.

2. **Overhead:** The combined time for Trajectory Fitting (pre-computation) and Inference (which includes CDE integration).

**Breakdown.** Standard NCDEs suffer from high training latency due to the stiffness of the control paths. In contrast, our smoothing-based methods incur a one-time pre-computation cost but drastically reduce the pure training time. This shift of complexity from the iterative loop to pre-processing results in a significantly lower total time to convergence. Figure 10 (bottom) shows that even for MVC-GP, preprocessing remains small relative to integration, consistent with the single-trajectory GP and kernel cases.

### C.4. Attention Head Diversity and Regularization

To investigate whether the multiple attention heads in our Multi-View architectures learn distinct temporal representations, we analyzed the diversity of the attention weights on the UWaveGestureLibrary dataset. We quantify this diversity using the maximum pairwise Cosine Distance between the attention score distributions.

Furthermore, we experimented with explicitly enforcing diversity during training by introducing a regularization term $\mathcal{L}_{\text{reg}}$ (with coefficient $\lambda = 0.1$) that penalizes high cosine similarity between heads. As demonstrated in Table 4, applying this regularization successfully decreases the maximum cosine similarity (i.e., increases diversity). However, it simultaneously degrades classification accuracy (e.g., from 92.05% to 87.50% for MVC GP). This reveals a critical trade-off: strictly forcing heads to look at different parts of the trajectory causes some heads to attend to uninformative noise simply to remain distinct. Our unregularized MVC-CDE naturally strikes an optimal balance, maintaining sufficient diversity (driven by heterogeneous bandwidth initialization) while enabling heads to collaborate on task-relevant features.

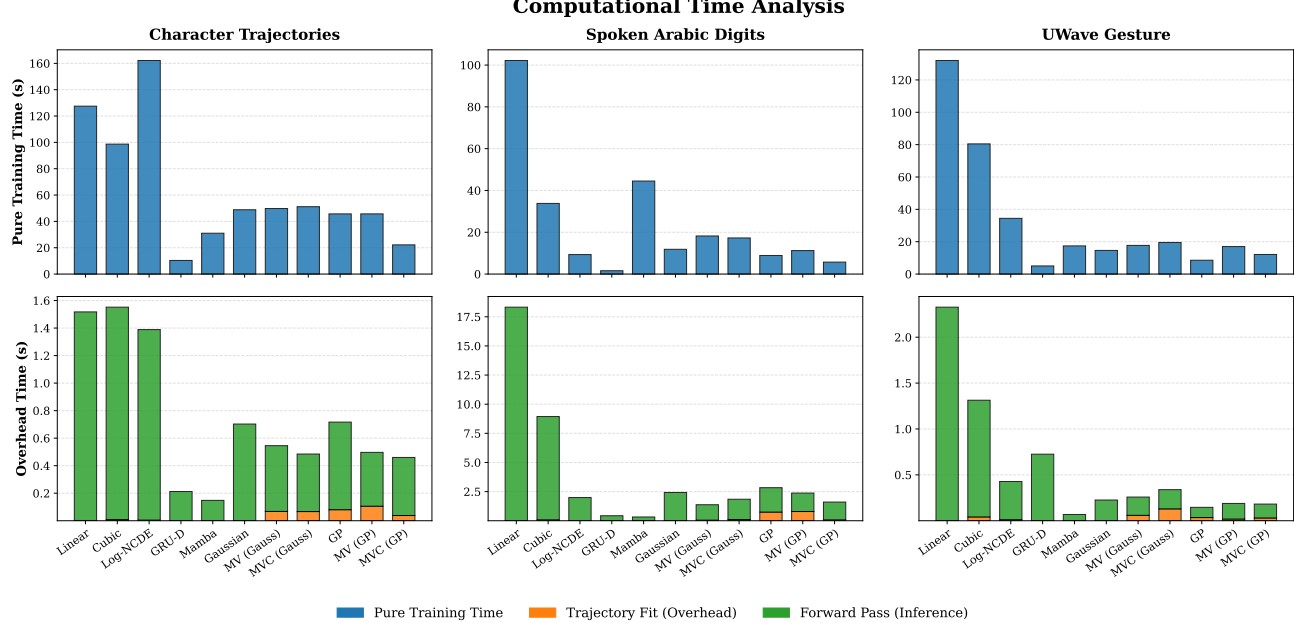

*Figure 10.* **Computational Cost Breakdown (NVIDIA H100, small batches).** Top: Pure training time significantly decreases for smoothing-based models compared to standard Linear and Cubic baselines, as the smoother paths reduce solver stiffness. Bottom: The overhead analysis shows that our method incurs a one-time trajectory-fitting cost (orange segments), which is outweighed by the substantial reduction in iterative training time.

*Table 4.* **Head Diversity and Regularization Impact (UWaveGestureLibrary).** Comparison of Maximum Cosine Similarity between attention heads and regularization effect on accuracy between MV and MVC models with 5 heads (Bandwidths: [0.03, 0.3, 1.4, 2.4, 4.0]), GP Kernel.

| Architecture | Max Cosine Similarity ↓ | Test Accuracy (%) ↑ |
|---|---|---|
| MV GP | $0.383 \pm 0.045$ | $86.36 \pm 1.45$ |
| MV GP + Regularization | $0.120 \pm 0.021$ | $80.68 \pm 1.82$ |
| **MVC GP (Ours)** | $\mathbf{0.222 \pm 0.033}$ | $\mathbf{92.05 \pm 1.12}$ |
| MVC GP + Regularization | $0.069 \pm 0.014$ | $87.50 \pm 1.55$ |

## C.5. Robustness to Irregular Sampling

In real-world scenarios, sensor failures or asynchronous recording often lead to missing data. To evaluate our model's robustness to sparse observational grids, we simulated missingness on the Speech Commands dataset by randomly dropping 30% and 50% of the available observations.

As shown in Table 5, baseline Spline CDEs suffer a noticeable performance drop and exhibit extremely high NFE. This occurs because exact deterministic interpolants create sharp, unnatural artifacts when spanning large gaps between missing observations, forcing the solver to take tiny steps. Conversely, MVC GP demonstrates strong robustness, maintaining over 89% accuracy even at a 50% drop rate, while keeping integration costs consistently low. The GP prior effectively imputes the missing gaps with smooth, well-behaved trajectories.

*Table 5.* **Robustness to Irregular Sampling (Speech Commands).** Evaluation under different data drop rates.

| Drop Rate | Model | Accuracy (%) ↑ | Avg NFE ↓ | Train Time (s) ↓ |
|---|---|---|---|---|
| 30% | Baseline (Cubic) | $73.31 \pm 2.10$ | $425 \pm 18$ | 16,131 |
| | Baseline (Linear) | $74.19 \pm 2.45$ | $838 \pm 25$ | 6,124 |
| | **MVC GP (Ours)** | $\mathbf{90.44 \pm 1.15}$ | $\mathbf{236 \pm 15}$ | $\mathbf{1,882}$ |
| 50% | Baseline (Cubic) | $73.44 \pm 2.41$ | $378 \pm 14$ | 14,127 |
| | Baseline (Linear) | $74.37 \pm 1.85$ | $743 \pm 20$ | 7,696 |
| | **MVC GP (Ours)** | $\mathbf{89.87 \pm 0.85}$ | $\mathbf{164 \pm 12}$ | $\mathbf{1,598}$ |

Furthermore, while computationally lightweight discrete baselines like GRU-D perform well on regularly sampled data, their performance degrades drastically when observations are systematically missing. To explicitly demonstrate this practical trade-off, we evaluated GRU-D against MVC GP under random observation drop rates of 30% and 50% across the classification benchmarks.

As shown in Table 6, MVC GP maintains a significant accuracy advantage. This confirms that the continuous-time representation and robust path smoothing justify the additional computational cost, since discrete RNNs fail to capture the underlying dynamics robustly in highly irregular regimes.

*Table 6.* **Comparison of Accuracy across irregular datasets.** Evaluation of GRU-D vs. MVC GP under 30% and 50% observation drop rates.

| Dataset | Model | Drop Rate = 0.3 | Drop Rate = 0.5 |
|---|---|---|---|
| | | Acc (%) | Acc (%) |
| CharacterTrajectories | GRU-D | $73.18 \pm 0.65$ | $72.45 \pm 0.82$ |
| | MVC GP (Ours) | $\mathbf{91.61 \pm 0.55}$ | $\mathbf{89.86 \pm 0.71}$ |
| SpokenArabicDigits | GRU-D | $92.95 \pm 0.24$ | $91.07 \pm 0.31$ |
| | MVC GP (Ours) | $\mathbf{97.33 \pm 0.35}$ | $\mathbf{96.59 \pm 0.42}$ |
| UWaveGestureLibrary | GRU-D | $75.59 \pm 1.85$ | $69.45 \pm 2.74$ |
| | MVC GP (Ours) | $\mathbf{86.36 \pm 0.84}$ | $\mathbf{84.09 \pm 1.12}$ |

## C.6. Impact of Adaptive ODE Solvers

The choice of the numerical ODE solver plays a crucial role in the speed-accuracy trade-off. We evaluated our method alongside baselines using solvers of varying mathematical orders: `bosh3` (Bogacki-Shampine, 3rd order), `dopri5` (Dormand-Prince, 5th order), and `dopri8` (8th order).

High-order solvers (like `dopri8`) assume the underlying vector field is highly smooth. If the driving path contains residual local variations, the strict local truncation error bounds of an 8th-order solver are frequently violated, leading to massive step rejections and a spike in NFE. This phenomenon is clearly visible in Table 8, where `dopri8` drastically increases

computational time. Conversely, because our GP smoothing produces well-regularized paths without extreme high-frequency jumps, lower-to-mid order solvers become highly efficient. For MVC GP, `bosh3` halves the NFE compared to `dopri5` with only a marginal drop in accuracy (Table 7), offering an excellent configuration for latency-critical applications.

*Table 7.* **Impact of Adaptive ODE Solvers on Accuracy (%).** Evaluation on the UWaveGestureLibrary dataset across different solver orders.

| Model | `dopri5` (Standard) | `bosh3` (Low-order) | `dopri8` (High-order) |
|---|---|---|---|
| Linear (NCDE) | $86.59 \pm 4.43$ | $85.12 \pm 2.80$ | $82.95 \pm 3.10$ |
| Cubic (NCDE) | $89.09 \pm 3.65$ | $88.64 \pm 2.15$ | $81.82 \pm 4.25$ |
| **MVC GP (Ours)** | $\mathbf{95.39 \pm 0.28}$ | $\mathbf{93.18 \pm 0.75}$ | $\mathbf{89.77 \pm 1.20}$ |

*Table 8.* **Impact of Adaptive ODE Solvers on Computational Efficiency.** Evaluation on the UWaveGestureLibrary dataset. Values are presented as Total Training Time (s) / Average NFE.

| Model | `dopri5` (Standard) | `bosh3` (Low-order) | `dopri8` (High-order) |
|---|---|---|---|
| Linear (NCDE) | $128 \pm 5$ / $2077 \pm 71$ | $320 \pm 14$ / $4150 \pm 115$ | $1103 \pm 45$ / $19672 \pm 415$ |
| Cubic (NCDE) | $75 \pm 3$ / $1075 \pm 78$ | $192 \pm 8$ / $2322 \pm 85$ | $278 \pm 11$ / $4449 \pm 125$ |
| **MVC GP (Ours)** | $\mathbf{12.29 \pm 0.27}$ / $218 \pm 10$ | $19.93 \pm 0.85$ / $\mathbf{108 \pm 6}$ | $25.51 \pm 1.15$ / $393 \pm 18.50$ |

## C.7. High-Dimensional Scalability

To test the limits of our architecture, we evaluated it on the highly multivariate PEMS-SF dataset, which contains 963 continuous feature dimensions. Processing continuous-time dynamics in such a high-dimensional space is notoriously slow for standard Neural CDEs.

Table 9 illustrates the classic trade-off between hardware-accelerated discrete models and continuous-time expressive models. Mamba, utilizing a highly optimized discrete parallel scan, trains exceptionally fast (41 seconds). However, it struggles to fully capture the complex, asynchronous spatial-temporal dependencies of the 963 sensors, achieving only 71.59% accuracy. Our MVC GP, while computationally heavier than Mamba, is nearly an order of magnitude faster than standard Linear CDEs (1,679s vs 13,049s) and achieves a dominant accuracy of 84.09%. This confirms that path smoothing makes continuous-time modeling highly tractable even for nearly 1000-dimensional systems. During inference on PEMS-SF, preprocessing adds minimal overhead relative to the full forward pass: trajectory fitting accounts for only 1.33% of total test time, while the combined encoder and ODE solve account for 98.67%. Smoothing the driving signal therefore accelerates integration without creating an inference-time bottleneck at this scale.

*Table 9.* **High-Dimensional Scalability (PEMS-SF dataset, 963 dimensions).**

| Model | Acc (%) ↑ | NFE ↓ | Train Time (s) ↓ |
|---|---|---|---|
| Baseline (Linear) | $63.64 \pm 0.9$ | $6383 \pm 159$ | 13,049 |
| Baseline (Cubic) | $67.05 \pm 1.4$ | $1163 \pm 97$ | 5,934 |
| Mamba | $71.59 \pm 1.1$ | - | **41** |
| **MVC GP (Ours)** | $\mathbf{84.09 \pm 0.8}$ | $\mathbf{225 \pm 12}$ | 1,679 |

## C.8. Comparison with Smoothing Splines

To ensure that our performance gains are not solely due to basic trajectory smoothing, we implemented a baseline using standard statistical Smoothing Splines. Specifically, we preprocessed the raw sequences using SciPy's `UnivariateSpline` applied independently to each spatial channel. To account for varying sequence lengths and missing data, the smoothing penalty $s$ was computed dynamically for each sequence as $s = \alpha \times N_{\text{valid}}$, where $N_{\text{valid}}$ is the number of valid (non-NaN) observations in the specific channel, and $\alpha = 0.5$ is a constant scaling hyperparameter. The resulting smoothed paths were then converted into standard natural cubic CDE coefficients.

As shown in Table 10, the Smoothing Spline baseline performs reasonably well but still falls short of our Multi-View architectures. The fundamental limitation of a traditional smoothing spline is its static, global nature: it applies a uniform degree of smoothing across the entire time series, which inevitably destroys localized high-frequency signals necessary for classification. Our MVC GP overcomes this by combining multiple heterogeneous smoothing scales with an attention mechanism, dynamically preserving task-relevant details while effectively suppressing solver-stiffening noise.

*Table 10.* **Accuracy Comparison (%) including Smoothing Spline Baseline.**

| Model | CharacterTrajectories | SpokenArabicDigits | UWaveGestureLibrary |
|---|---|---|---|
| Linear (NCDE) | $90.59 \pm 1.47$ | $89.44 \pm 0.93$ | $86.82 \pm 4.22$ |
| Cubic (NCDE) | $91.89 \pm 0.47$ | $88.70 \pm 1.03$ | $85.68 \pm 3.47$ |
| Smoothing Spline | $90.80 \pm 0.89$ | $96.81 \pm 0.34$ | $85.45 \pm 4.85$ |
| MVC Gaussian (Ours) | $91.71 \pm 1.40$ | $97.34 \pm 0.46$ | $90.00 \pm 2.03$ |
| **MVC GP (Ours)** | $\mathbf{95.07 \pm 0.49}$ | $\mathbf{98.28 \pm 0.29}$ | $\mathbf{95.39 \pm 0.28}$ |

