# OpenReview forum: "Efficient Neural Controlled Differential Equations via Attentive Kernel Smoothing"
_ICML.cc/2026/Conference — ICML 2026 regular_

### Official Review · Reviewer_2ruU · 2026-02-28

**Soundness:** 3
**Presentation:** 3
**Significance:** 3
**Originality:** 3
**Overall Recommendation:** 4
**Confidence:** 4

**Summary:**

This paper proposes a smoothing-based framework to reduce the Number of Function Evaluations (NFE) in Neural CDEs. The core idea is to replace exact spline interpolation with kernel or Gaussian Process smoothing, and to compensate for information loss via a multi-head attention-based Multi-View CDE (MV-CDE) architecture.

This submission’s major idea concerns decoupling numerical stiffness from input noise by explicitly controlling path regularity before ODE integration. The work assesses the concept that smoothing the control path—combined with attentive recovery—can significantly reduce solver complexity while preserving predictive performance. The paper is theory-oriented and provides substantial analytical development, including formal relationships between higher-order derivatives of the control path and solver complexity, as well as explicit NFE scaling laws under smoothing.

**Compliance With Llm Reviewing Policy:**

Affirmed.

**Final Justification:**

I recommend a weak accept (4). The paper presents strong theoretical development, with clear derivations linking higher-order smoothness to solver efficiency, and a well-motivated, theory-driven design. The empirical results are also convincing, showing substantial speedups while maintaining accuracy.

My main concerns relate to clarity and completeness of the theoretical presentation. In particular, the definition of NFE is not formally specified, the role of bandwidth selection is not fully discussed, and the dependence on solver tolerance is not explicitly characterized. Some notational issues could also be improved for precision. The empirical evaluation, while solid, would benefit from validation on more challenging real-world settings.

The rebuttal addressed several of these concerns and improved clarity, although some points could still be strengthened. Overall, I find the contribution solid and technically meaningful, and I am comfortable recommending acceptance.

**Key Questions For Authors:**

1.	Missing formal definition of NFE.

Before Theorem 3.1, NFE is described only informally. However, the theorem states

$$
\mathrm{NFE} \propto \int_0^T \left| X^{(p+1)}(t) \right|^{\frac{1}{p+1}} dt,
$$

without clearly specifying whether NFE refers to total function evaluations (including rejected steps), number of accepted steps, or total calls to f_\theta. Since the theoretical scaling depends on what is being measured, a precise mathematical definition would improve rigor.

2.	Incomplete discussion of bandwidth selection.

For the Nadaraya–Watson estimator, classical nonparametric theory suggests that the optimal bandwidth scales as

$$
h \sim n^{-1/5},
$$

or can be selected via cross-validation. The paper does not discuss how the proposed bandwidth strategy compares with classical bias–variance optimality, nor how statistical optimality trades off against computational efficiency (NFE reduction). Clarifying this trade-off would strengthen the theoretical positioning.

3.	Missing explicit dependence on global error tolerance.

Theorem 3.4 states

$$
\mathrm{NFE}_{\text{total}} \propto \left( \min_m h_m \right)^{-1},
$$

but suppresses the dependence on the solver tolerance \delta. Since Theorem 3.1 explicitly links step size to \delta, it would be important to characterize how NFE scales jointly with smoothing parameter and tolerance.

4.	Notational precision.

The symbol “∝” is repeatedly used for scalar quantities. In several places, a bound such as

$$
\mathrm{NFE} \lesssim  h^{-1}
$$

(where \lesssim denotes inequality up to a positive constant) would be mathematically more precise.

5.	Limited real-world validation.

While the benchmarks are standard time-series datasets, they remain relatively controlled classification tasks. Demonstrating effectiveness on more challenging real-world irregular time-series problems would strengthen the empirical impact.

If the authors can provide clear and complete responses to the above questions, I would be willing to increase my score accordingly.

**Limitations:**

See Key Questions For Authors.

**Strengths And Weaknesses:**

1.	Strong theoretical development.
The paper provides detailed derivations connecting higher-order derivatives of the control path to adaptive solver step size and NFE (Theorem 3.1 and Appendix A). The comparison between deterministic splines and smoothing-based constructions is rigorous and well motivated.
2.	Theory-driven design.
The architectural choices (multi-view smoothing, bandwidth control) are clearly inspired by the theoretical scaling laws. In particular, the result
\mathrm{NFE}(X_h) \propto h^{-1}
provides a clean explanation of the computational–regularity trade-off.
3.	Empirical validation.
The experiments show consistent speedups (4×–14×) while preserving or improving accuracy across benchmarks.

---

> ### Author Rebuttal · Authors · 2026-03-30
>
> **1\. Mathematical Definitions and Notation**\
> We will update the manuscript to incorporate your feedback on formal precision.
>
> **1\.1 (NFE Definition):** We clarify that NFE refers strictly to the total number of calls to the vector field, including both accepted and rejected steps during adaptive integration.
>
> **1\.2 Tolerance & Bandwidth in Theorem 3.4:** Theorem now incorporates the solver tolerance $\delta$. We propagated $\delta$ from Theorem 3.1 through the multi-head integration. Applying the smoothing bound $\| \mathbf{X}^{(p+1)} \| \lesssim h^{-(p+1)}$ to the local truncation error condition, the step size constraint for the $m$\-th head becomes:
>
> $$
> \eta_i^{(m)} \lesssim \left( \frac{\delta}{C} \cdot h_m^{p+1} \right)^{\frac{1}{p+1}} = C^{-\frac{1}{p+1}} \delta^{\frac{1}{p+1}} h_m \lesssim \delta^{\frac{1}{p+1}} h_m
> $$
>
> To satisfy the global tolerance $\delta$ across all $M$ heads simultaneously, the stiffest path dictates the global step size:
>
> $$
> \eta_i \lesssim \delta^{\frac{1}{p+1}} \min_{m=1}^M h_m
> $$
>
> Integrating over the interval $[0, T]$ yields the joint scaling law for the total NFE:
>
> $$
> NFE_{\text{total}} \lesssim \delta^{-\frac{1}{p+1}} \left( \min_{m=1}^M h_m \right)^{-1}
> $$
>
> **1\.3 Notational precision, \\propto vs \\lessim:** We have updated all relevant scalar quantity bounds, replacing \\propto with \\lesssim to reflect inequality up to a positive constant.
>
> **2\. Bandwidth Selection and Bias-Variance Trade-off**
>
> We thank the reviewer for raising this important point. While classically optimal bandwidths ($h \propto n^{-1/5}$) minimize Asymptotic Mean Integrated Squared Error (AMISE) via curve-level cross-validation, Neural CDEs introduce a distinct three-way trade-off between statistical bias, variance, and computational efficiency (NFE).
>
> Classically optimal bandwidths retain high-frequency local fluctuations. While optimal for pure curve reconstruction, these fluctuations force adaptive ODE solvers to take excessively small steps, creating a computational bottleneck. Consequently, classical statistical optimality effectively trades off poorly against NFE reduction.
>
> To resolve this, our approach treats it as a task-aware hyperparameter tuned on a downstream validation set. This setup natively favors a larger bandwidth ($h > c \cdot n^{-1/5}$), deliberately oversmoothing the path (increasing statistical bias) to dramatically lower NFEs. Ordinarily, this oversmoothing would hurt predictive performance by obscuring high-frequency features. However, our proposed Attention-based Multi-View CDE (MVC-CDE) explicitly mitigates this penalty; its learnable queries recover the task-relevant details lost to smoothing.
>
> We will add a dedicated subsection to the paper, exploring this trade-off between classical bandwidth selection, task-aware tuning, and ODE solver efficiency.
>
> **3\. Real-World Validation**\
> To address concerns about limited real-world validation, we expanded our empirical evaluation to include forecasting tasks and irregularly sampled data, testing the model in less controlled environments.
>
> **3\.1 Forecasting Under Extreme Missingness.** We added a minimal multivariate forecasting experiment on the real-world ETTm1 dataset, where robust handling of asynchronous sampling and missingness is essential. We stress-tested our continuous-time model by simulating extreme sensor failure: dropping 30% of the daily blocks. Using a 24-step history to forecast the next 24 steps, we evaluate performance on non-controlled, causally structured tasks--exactly where impute-then-analyze pipelines often fail. For a quick baseline check, we compare MVC against Neural CDE and a standard GRU. All models use history-to-embedding encoding followed by a lightweight linear projection to predict all future steps at once (DLinear).
>
> **We provide our results in Table 1:** [https://bashify.io/i/gbPV3R#](https://bashify.io/i/gbPV3R#)
>
> As you can see, our model comfortably outperforms competitors.
>
> **3\.2 Irregular Setting:** We added a new Speech Commands dataset <https://doi.org/10.48550/arXiv.1804.03209>. We applied distinct data drop rates to test scalability and evaluate performance on missing values.
>
> **We provide our results in Table 2:** [https://bashify.io/i/mRPGPz](https://bashify.io/i/mRPGPz)
>
> MVC GP maintains accuracy on irregular data. Sparse inputs trigger more solver steps in Cubic and Linear baselines. These models require frequent function evaluations and longer training times. GP kernel smoothing provides a stable path for the solver when observations are missing. The model keeps the number of function evaluations low in these sparse settings.

---

> > ### Author Rebuttal · Reviewer_2ruU · 2026-04-01
> >
> > Thank you for your response. I will improve my score.

---

> > > ### Author Response · Authors · 2026-04-07
> > >
> > > Thank you for your positive feedback!

---

### Official Review · Reviewer_f8og · 2026-03-05

**Soundness:** 3
**Presentation:** 3
**Significance:** 2
**Originality:** 3
**Overall Recommendation:** 5
**Confidence:** 4

**Summary:**

This work aims to reduce the number of function evaluations (NFEs) in Neural CDEs that arise from the roughness of the driving signal. The authors address this issue by applying smoothing techniques (Kernel smoothing and Gaussian Processes) to control the regularity of the driving signal. To mitigate potential information loss caused by smoothing, they introduce MV-CDE and MVC-CDE, which incorporate multiple smoothed versions of the driving signal simultaneously. Experimental results show that the proposed methods achieve strong performance while significantly reducing computational cost across several tasks.

**Compliance With Llm Reviewing Policy:**

Affirmed.

**Key Questions For Authors:**

- Several figures refer to “cubic.” It appears that this corresponds to cubic spline interpolation, is that correct? If it indeed refers to cubic spline interpolation, it would also be helpful to include comparisons with other smoothing techniques, such as cubic spline regression, and discuss why the proposed kernel- and GP-based smoothing approaches are preferable in this setting.

**Limitations:**

Limitations are not discussed. Please discuss limitations mentioned in the weakness part.

**Strengths And Weaknesses:**

**Strength:**

- **Insightful idea.** The key insight of using smoothed driving signals to reduce the NFEs required by adaptive ODE solvers is interesting and well-motivated.
- **Strong theoretical presentation.** The paper provides solid theoretical analysis, and the overall presentation is clear and easy to follow.
- **Solid experimental evaluation.** The experiments are comprehensive and convincingly demonstrate the effectiveness and performance improvements of the proposed methods.

**Weakness:**

- **Sensitivity to hyperparameters.** The proposed method depends on two important hyperparameters: the number of heads and the smoothing bandwidth. Tuning these parameters may require considerable effort. Although the error rate generally decreases as the number of heads increases, the method remains highly sensitive to the choice of bandwidth, and the paper provides limited guidance on how to select an appropriate value. This issue is also common in kernel-based methods.
- **Limited applicability to solver types.** The approach appears to be designed specifically for **adaptive ODE solvers** and may not be applicable to **fixed-step solvers**. This limitation should be discussed more clearly in the paper.
- **Evaluation on additional adaptive solvers.** The experiments primarily use the **Dormand–Prince solver**. It would strengthen the empirical evaluation to test the method on other adaptive ODE solvers to better demonstrate the generality of the proposed smoothing strategy for reducing NFEs in Neural CDEs.
- **Potential scalability concerns in high dimensions.** While the smoothed driving signal helps reduce NFEs and thus accelerates computation, the multi-view design introduces additional cost. Specifically, when using $N$ heads, the model requires $N$ matrix–vector multiplications, but standard Neural CDE only require one. For extremely high-dimensional systems, this overhead could become a bottleneck, potentially reducing or even negating the overall speedup compared to standard Neural CDEs.

---

> ### Author Rebuttal · Authors · 2026-03-30
>
> **1\. Hyperparameter Selection**
>
> **1\.1 Bandwidth Selection**
>
> Classical bandwidth selection $h \sim n^{-1/5}$ (where $n$ is the sample size) minimizes curve reconstruction error. Neural CDEs require a different balance between statistical bias, variance, and computational efficiency (NFE). Statistically optimal paths often retain high-frequency noise. This noise forces adaptive ODE solvers to take smaller steps, which increases NFE.
>
> We tune h on a validation set to prioritize task-aware efficiency. This process favors larger bandwidths that oversmooth the trajectory. While oversmoothing increases statistical bias, it significantly reduces NFE. MVC-CDE recovers the details lost during smoothing through its multi-view mechanism. We added a subsection to the paper that explores this mathematical trade-off among classical selection, validation tuning, and solver efficiency.
>
> **1\.2 Head Count** Figure 6 in our article shows the error rate as a function of head count M. We found that M = 4 provides the best balance between model capacity and performance across all tested datasets. Because the smallest bandwidth in the multi-view set determines the global integration speed, we select this minimum value based on the NFE-accuracy trade-off. Larger bandwidths in the remaining heads provide the necessary regularization without slowing the solver.
>
> **2\. Solver Types and Dopri5** Indeed, our method requires the use of adaptive-step solvers; we will highlight this more clearly in the paper’s text and add it to the limitations section. Thank you for pointing this out. As for our specific solver choice, we settled on the `dopri5` method, since it serves as the default option in modern ODE solvers, including the most popular NeuralODE and Neural CDE libraries, torchdiffeq and torchcde; the less popular torchode library; scipy’s solve_ivp method, and matplotlib’s ode45.
>
> We further validate the efficiency of the dopri5 method, comparing it to two other popular solvers: the dopri8 method (of higher order) and the bosh3 method (of lower order). We report the resulting Accuracy and Timing results separately.
>
> **We provide the performance of different adaptive ODE solvers on the UWaveGestureLibrary dataset (Accuracy, %) in** **Table 1:** [https://bashify.io/i/hx7X50](https://bashify.io/i/hx7X50)**, and the respective time statistics in** **Table 2:** [https://bashify.io/i/xH6fQM](https://bashify.io/i/xH6fQM)**.**
>
> As you can see, Dopri 5 is both the fastest and the highest-performing option. In line with prior work, this solver strikes the perfect balance between efficiency and precision. We will include this table and the corresponding discussion in the Appendix of the revised manuscript to explicitly address the performance on various adaptive solvers.
>
> **3\. High-Dimensional Scalability**
>
> We evaluated MVC GP on the PEMS-SF dataset (963 dimensions). Using \$M\$ heads increases the computational cost per ODE step. Reducing the total number of function evaluations (NFEs) offsets this penalty. MVC GP with 3 heads requires fewer NFEs than the cubic and linear baselines. This cuts training time and improves accuracy on high-dimensional data.
>
> **We provide the results in** **Table 3:** [https://bashify.io/i/bQSzsB](https://bashify.io/i/bQSzsB)**.**
>
> During inference, preprocessing adds minimal overhead. The MVC GP model computes the forward pass and solves the ODE, accounting for 98.67% of the total test time. Fitting the interpolation takes 1.33%. Smoothing the driving signal accelerates the system without creating bottlenecks. We will add these PEMS-SF results to the revision.
>
> **4\. Cubic Interpolation vs. Spline Regression** Indeed, "cubic" refers to natural cubic spline interpolation, which passes through all observations. We also evaluated Smoothing Splines (cubic spline regression) as an alternative baseline.
>
> **We provide accuracy comparison (%), including the new Smoothing Spline baseline in** **Table 4:** [https://bashify.io/i/S30TIL](https://bashify.io/i/S30TIL)**, while the respective Time / NFE comparison is given by** **Table 5:** [https://bashify.io/i/gkwHZM](https://bashify.io/i/gkwHZM)**.**
>
> Smoothing Splines reduce NFE but our multi-view approach achieves higher accuracy by aggregating distinct temporal scales. Unlike smoothing splines that apply global second-derivative regularization, kernel and GP methods use local spatial filters. GPs handle heteroscedastic noise through their probabilistic formulation, enabling our multi-view attention to dynamically scale observation weights. That said, cubic spline smoothing could also be integrated into our MVC framework as an alternative kernel.
>
> **5\. Limitations:** we will add a dedicated limitations section, highlighting the aforementioned solver applicability and the necessity of bandwidth tuning.

---

> > ### Author Rebuttal · Reviewer_f8og · 2026-03-31
> >
> > My concerns are fully solved and I will raise my score. But I would appreciate it if the authors can provide me with detailed reference for the classical bandwidth selection $h \sim n^{-1/5}$.

---

> > > ### Author Response · Authors · 2026-04-07
> > >
> > > Thank you for the positive feedback! The classical bandwidth selection formula is discussed in, for instance, Silverman's "Density Estimation for Statistics and Data Analysis" (1986). See Section 3.4, specifically pages 45–47.

---

### Official Review · Reviewer_76S8 · 2026-03-07

**Soundness:** 3
**Presentation:** 4
**Significance:** 3
**Originality:** 3
**Overall Recommendation:** 4
**Confidence:** 4

**Summary:**

This paper addresses a fundamental computational bottleneck in Neural Controlled Differential Equations (Neural CDEs): the excessively high Number of Function Evaluations (NFE) triggered by the roughness of standard spline-interpolated control paths. To alleviate solver stiffness, the authors propose replacing exact interpolation with Kernel and Gaussian Process (GP) smoothing. To counteract the inevitable loss of high-frequency information caused by smoothing, they introduce an attention-based Multi-View CDE (MVC-CDE) architecture that dynamically aggregates representations from multiple trajectories with varying bandwidths. Empirical evaluations demonstrate substantial reductions in NFE and training time, alongside competitive or improved predictive accuracy.

**Compliance With Llm Reviewing Policy:**

Affirmed.

**Final Justification:**

The rebuttal addresses several concerns with additional experiments and clarifications. However, the gains remain relatively modest given the added complexity, and the contribution appears incremental. I therefore maintain my score.

**Key Questions For Authors:**

1. Streaming Latency: Can you provide a wall-clock latency profiling for an online streaming inference scenario? Specifically, how computationally prohibitive is the trajectory smoothing step when new observations must be processed sequentially in real-time?

2. Memory Overhead & Alignment: What is the exact memory overhead (VRAM) of MVC-CDE compared to a standard single-path Neural CDE? Could you provide an ablation study comparing MVC-CDE with a single-path model under a strictly matched parameter and memory budget?

3. SSM Comparisons: How does the proposed framework compare in terms of throughput (sequences/second) and accuracy against solver-free continuous-time models like S4 or Mamba?

**Limitations:**

While the appendix reports trajectory fitting overhead, the paper does not clearly discuss how this preprocessing cost may affect deployment in streaming or low-latency settings.

**Strengths And Weaknesses:**

Strengths

1. Highly Practical Motivation: Targeting the geometric roughness of the driving path as the root cause of ODE solver stiffness is an insightful and highly practical diagnosis of a known systemic bottleneck in Neural CDEs.

2. Elegant Architectural Compromise: The use of an attention-based multi-view mechanism to fuse multi-scale smoothed dynamics is an intuitive and effective architectural choice to balance trajectory regularity with the retention of high-frequency details.

3. Significant Empirical Gains: The reported reductions in NFE and the corresponding speedups in pure training time are highly compelling and thoroughly validated across multiple diverse benchmarks.

Weaknesses

1. The "Preprocessing Trap" in Streaming Inference: While the approach successfully reduces the ODE solver's integration time, it introduces a non-trivial preprocessing overhead (fitting GPs or Kernels for the trajectories) . For offline training, this one-time cost is perfectly acceptable. However, for online or streaming inference scenarios (e.g., real-time medical monitoring or high-frequency trading), recomputing or dynamically updating the smoothed paths as new observations arrive sequentially could become a significant latency bottleneck. The paper severely lacks a critical discussion on this operational constraint.

2. Unclear Pareto Frontier for the Multi-View Mechanism: MVC-CDE requires solving multiple smoothed trajectories simultaneously. It remains unclear whether solving an ensemble of smoothed CDEs strictly provides a better compute-accuracy Pareto frontier than simply running a single CDE with an augmented hidden state dimension. The evaluation lacks a rigorous baseline comparison where the total parameter count and peak memory footprint (VRAM) are strictly matched.

3. Absence of Modern Continuous-Time Baselines: The sequence modeling landscape has recently been revolutionized by State Space Models (e.g., S4, Mamba) that bypass explicit ODE solvers entirely via hardware-efficient discrete operators. A discussion—and ideally an empirical comparison—detailing how this accelerated Neural CDE framework conceptually and practically competes with these inherently faster continuous-time counterparts would help better contextualize the contribution.

4. Conceptual tension with the causal motivation of Neural CDEs
Neural CDEs are often motivated by their ability to naturally handle irregularly sampled and continuously arriving data streams. The proposed global smoothing procedures (especially GP smoothing) appear to assume access to the full trajectory when constructing the control path. While this is reasonable for offline sequence classification benchmarks, it partially weakens the causal/streaming interpretation that originally motivated Neural CDEs. A clearer discussion of this trade-off between computational efficiency and causal modeling would strengthen the paper.

---

> ### Author Rebuttal · Authors · 2026-03-30
>
> **1\. Streaming Latency and Preprocessing Overhead.** We recognize the preprocessing cost in online streaming settings. Thank you for this comment. In the submitted manuscript, we present the preprocessing overhead timing in Figure 10, in the Appendix of our paper, comparing the time for a single forward pass to the fitting time. We point out the following observations:
>
> -  For single-trajectory Gaussian Kernel smoothing, the interpolation cost is negligible.
>
> -  For MV(C) Gaussian Kernel smoothing, and for single-trajectory GP, the interpolation cost is quite small, smaller or comparable to the cost of integration.
>
> -  However, MV and MVC GP models incur significant fitting overheads.
>
> Regarding the final point: the per-trajectory fitting process is parallelizable, so that overhead is due to us hitting the ceiling in the number of parallel computations; otherwise, the cost would be pretty much comparable to that of a single-trajectory GP. Indeed, in this experiment, we processed the input with a very large batch size, which caused the GPU’s parallelization capacity to reach its limit. To validate this claim, we re-run this image on a more capable GPU (H100) with a small batch size--the latter better aligns with online tasks.
>
> The results can be found in the following image: <https://bashify.io/i/rGrRQq>. Overall, given a small batch size and a capable GPU, even the most preprocessing-heavy model, MVC-GP, does not incur a significant overhead, so it can be used in a streaming setting as-is.
>
> **2\. Memory Alignment and SSM Baselines ([Mamba](https://doi.org/10.48550/arXiv.2312.00752))** We matched parameter counts across standard NCDEs (Cubic and Linear splines), our MVC architecture, and a pure-PyTorch Mamba implementation, by augmenting the number of hidden channels. The following tables detail Peak VRAM, parameter count, total training time, and accuracy. We did not strictly match VRAM: one can either equate parameter counts or memory footprints, but not both; we chose to match parameters, as this provides a standard, mathematically fair comparison of theoretical model capacity rather than forcing the single-path baseline into an artificial, over-parameterized regime just to consume equivalent memory.
>
> **We provide a comparison of model performance, given as A / B, where A is Accuracy (%), and B is total Training time (s.) in Table 1:**  <https://bashify.io/i/fEgR4b#>
>
> **And a comparison of Model Performance, Parameter Counts, and Peak Memory. Presented as A / B, where A is the number of parameters, and B is the peak VRAM in MB, in Table 2:** [https://bashify.io/i/llV9C4#](https://bashify.io/i/llV9C4#)
>
> MVC-CDE consumes more VRAM than standard NCDEs because it simultaneously integrates multiple paths. We emphasize that our Mamba baseline utilized a pure-PyTorch implementation. Consequently, it lacks the highly optimized, hardware-aware custom CUDA kernels.  However, it requires a fraction of the memory Mamba uses while delivering competitive or superior accuracy on these benchmarks. Mamba catches up to our model only on Characteristic Trajectories; the other benchmarks still favour MVC-CDE GP.
>
> **4\. Causal Motivation** We agree that Global GP/Kernel smoothing is an offline preprocessing step, trading strictly online/sequential processing for efficiency. However, as noted by [Morrill et al.](https://arxiv.org/abs/2106.11028), most standard Neural CDE implementations inherently rely on non-causal interpolations, such as cubic splines, making them theoretically unsuited for strict online streaming. Our primary contribution addresses the orthogonal challenge of computational efficiency: we regularize the control path to significantly reduce the Number of Function Evaluations (NFE) and inference time. We will clarify this distinction in a new "Online vs. Offline Processing Strategies" section.
>
> That said, to better address your concerns, we additionally include a strictly causal forecasting experiment on ETTm1, demonstrating how one can use such offline models in an inherently online task. In this setup, we forecast a 24-step horizon based on a 24-step history, and to challenge the true capability of continuous-time irregular models, we simulate extreme sensor failure by completely dropping 30% of the daily blocks. We do not precompute the interpolations: we fit them anew for each batch. For a quick baseline check, we compare MVC against Neural CDE and a standard GRU. To isolate the performance of the tested models and minimise the effect of the forecasting head, all models are used for history-to-embedding encoding, followed by a lightweight linear projection to predict all future steps at once (DLinear).
>
> **We provide an ETTm1 Forecasting Results** **in Table 3:** [https://bashify.io/i/lpn6BL#](https://bashify.io/i/lpn6BL#)
>
> As you can see, MVC-CDE comfortably outperforms competitors.

---

> > ### Author Rebuttal · Reviewer_76S8 · 2026-04-03
> >
> > Thank you for the detailed and thoughtful response. The clarifications regarding preprocessing overhead, streaming settings, and additional experiments are helpful and address several of my concerns.
> >
> > Overall, I maintain my score, as I view the contribution as a solid but incremental improvement to Neural CDE efficiency. In particular, while the method improves over spline-based CDEs, the gain over simpler baselines such as GRU appears relatively moderate, especially given the higher computational cost, which raises some questions about the practical trade-offs.

---

> > > ### Author Response · Authors · 2026-04-07
> > >
> > > Thank you for your constructive feedback.
> > >
> > > We acknowledge the computational trade-off between our method and simpler discrete baselines like GRU-D: GRU-D is computationally lighter, and the performance margin is modest. That said, the primary advantage of most continuous-time models over traditional Neural Networks lies in the robustness to severe data irregularity and missingness -- a critical factor in real-world scenarios where simple RNNs often fail. Consequently, despite increased compute requirements, Neural CDEs are still regarded as a prime choice for time series analysis, with new influential papers coming out regularly in major conference procedings \[1-6\].
> > >
> > > To explicitly demonstrate this practical trade-off, we evaluated both models on the discussed UEA datasets under random observation dropping (Drop Rates of 0.3 and 0.5). As shown in the table below, the accuracy of GRU-D degrades drastically when data is missing.
> > >
> > > Table: <https://bashify.io/i/cHxjib>
> > >
> > > We believe this substantial gain in robustness and accuracy justifies the increase in computational cost over discrete baselines. We will gladly include this table and a detailed discussion of these trade-offs in the final manuscript to provide a clearer picture for future readers.
> > >
> > > References:
> > >
> > > \[1\]: Kidger, P., et al. "Neural controlled differential equations for irregular time series." NeurIPS, 2020.
> > >
> > > \[2\]: Morrill, J., et al. "Neural rough differential equations for long time series." ICML, 2021.
> > >
> > > \[3\]: Chen, Y., et al. "Contiformer: Continuous-time transformer for irregular time series modeling." NeurIPS, 2023.
> > >
> > > \[4\]: Walker, B., et al. "Log Neural Controlled Differential Equations: The Lie Brackets Make A Difference." ICML, 2024.
> > >
> > > \[5\]: Majumdar, A., et al. "Neural Wave Equation for Irregularly Sampled Sequence Data." ICLR, 2025.
> > >
> > > \[6\]: Kuleshov, I., et al. "DeNOTS: Stable Deep Neural ODEs for Time Series." ICLR, 2026.

---

### Official Review · Reviewer_k4zT · 2026-03-13

**Soundness:** 3
**Presentation:** 3
**Significance:** 3
**Originality:** 3
**Overall Recommendation:** 5
**Confidence:** 3

**Summary:**

This paper studies the computational inefficiency of Neural CDEs and argues that a major source of cost is the roughness of the control path induced by exact interpolation of noisy observations. To address this, the paper replaces standard spline-based interpolation with smoother control paths constructed via kernel smoothing or Gaussian-process smoothing. Since naive smoothing can remove useful information, the paper further introduces a multi-view design in which multiple attention-guided smoothed paths are constructed and integrated jointly through a block-diagonal Neural CDE system. A convolutional front-end is also added in the MVC-CDE variant to provide richer local temporal features for attention. The paper evaluates the method on several UEA multivariate time-series classification datasets and reports strong accuracy together with substantial reductions in training time and number of function evaluations.

**Compliance With Llm Reviewing Policy:**

Affirmed.

**Final Justification:**

Thanks for the response, and my concerns have been adequately addressed.

**Key Questions For Authors:**

See questions in "Weakness" above, especially:
1.	A key part of the paper’s interpretation is that different heads learn complementary temporal views. What mechanism, beyond independent parameterization, is expected to prevent the heads from collapsing to similar solutions?
2.	Since the joint integration cost is limited by the stiffest head, how should one think about scaling the number of heads or including very fine smoothing scales? Is there a principled way to balance diversity against this efficiency bottleneck?
3.	Could you more explicitly quantify how much of the final gain comes from smoothing alone versus the multi-view mechanism versus the convolutional front-end?

**Limitations:**

See "Weakness" above.

**Strengths And Weaknesses:**

# Strength:
1.	The paper identifies a concrete and meaningful bottleneck in Neural CDEs.
Rather than treating efficiency as a generic solver issue, the paper points specifically to the roughness of the interpolated input path as the source of excessive small solver steps and high NFE. This is a clean and convincing starting point.
2.	The proposed solution is coherent and directly aligned with the stated problem.
The core idea is simple but well motivated: smoother control paths should reduce solver cost, but smoothing can lose information, so the model constructs multiple smooth views and combines them instead of relying on a single smoothed trajectory. This problem–method–tradeoff structure is easy to follow and makes sense.
3.	The empirical efficiency/accuracy tradeoff is appealing.
For a paper whose main selling point is computational efficiency, it is a strength that the experiments report not only accuracy but also runtime and NFE. The paper appears to provide a practically meaningful speedup without paying a major accuracy penalty.

# Weakness:
1.	The multi-view diversity story is intuitively appealing, but not fully convincing mechanistically.
A central part of the paper’s motivation is that different heads / paths can capture different temporal views and thereby compensate for information loss caused by smoothing. However, it is not clear what actually guarantees such diversity. The heads have separate parameters and can in principle learn different attention patterns, but there does not appear to be an explicit diversity-promoting mechanism that would prevent them from collapsing toward similar solutions. As a result, part of the observed gain may simply come from increased capacity or redundancy, rather than from genuinely complementary temporal views. I would have liked either a stronger empirical analysis of head diversity or some clearer discussion of why meaningful specialization should emerge reliably.
2.	The computational story is slightly weakened by the stiffest-head bottleneck.
The paper itself notes that, when multiple paths are integrated jointly, the global step size is constrained by the stiffest head, i.e. effectively by the smallest smoothing scale among them. This makes sense, but it also means that the multi-view mechanism has an inherent efficiency ceiling: once one head keeps a relatively fine-scale path, the whole joint integration pays for it. This does not invalidate the method, but it does temper the strength of the scalability story.
3.	It remains somewhat unclear how much of the gain comes from each component.
The paper compares several variants, which is helpful, but I still would have liked a cleaner decomposition of the final gain: how much comes from smoothing alone, how much from having multiple views, how much from GP-based smoothing rather than kernel smoothing, and how much from the convolutional front-end. The overall story is plausible, but a sharper component-level breakdown would make the paper stronger.
4.	The experimental scope is somewhat narrow relative to the broad efficiency motivation.
The motivation is presented fairly broadly as improving Neural CDE efficiency, but the experiments are focused on a small set of UEA multivariate time-series classification tasks. These are reasonable benchmarks, but the claim would feel stronger if the paper included at least one additional setting where Neural CDEs are also widely used.

---

> ### Author Rebuttal · Authors · 2026-03-30
>
> **1\. Head Diversity and Query Collapse** Architectures with fixed attention queries are prone to head collapse and redundancy. Prior work mitigates this using dynamic, input-dependent queries, often alongside regularization ([Shah et al., CVPR 2024](https://openaccess.thecvf.com/content/CVPR2024/papers/Shah_LQMFormer_Language-aware_Query_Mask_Transformer_for_Referring_Image_Segmentation_CVPR_2024_paper.pdf)). Adding the convolutional component (MVC-CDE) achieves this organically. Because convolutional filters capture varying local frequencies and temporal patterns, they provide a structural prior that directs attention heads toward distinct features, naturally preventing collapse. To validate this, we empirically compared the weight diversity of MV and MVC-CDE. Furthermore, we experimented with adding a regularization term to the loss function to penalize the maximum pairwise cosine similarity between attention heads.
>
> **We provide our results in Table 1**: <https://bashify.io/i/YMCvnX>
>
> The regularization reduces head similarity but lowers accuracy, suggesting over-constraint forces heads toward suboptimal features or noise. The MVC architecture's inherent diversity provides the best balance without regularization. Additionally, MVC achieves lower inter-head similarity than MV, indicating convolutions enhance diversity by offering richer context for attention heads.
>
> **2\. The Stiffest Head Bottleneck** We agree that the stiffest head dictates the global integration step, which creates an efficiency ceiling. We will explicitly discuss this in a dedicated Limitations section. Ultimately, this is a fundamental trade-off between regularity (smoothness) and fidelity (information retention). To balance diversity against this bottleneck in a principled way, we treat the smoothing bandwidths as hyperparameters selected on a log-scale, and we advise against including very fine scales. Empirically, MVC-CDE does not require fine-scale paths to achieve high accuracy. Across our hyperparameter searches, the method consistently favors and performs best with coarser smoothing scales (bandwidths > 1, typically ranging from 1 to 100). By anchoring our log-spaced bandwidths in this coarser regime, the multi-view mechanism successfully captures diverse temporal dynamics without ever triggering the extreme integration costs of a "stiff" path.
>
> **3\. Component-Level Breakdown**
>
> **3\.1. Smoothing (Gaussian Kernel and GP CDE):** Gaussian kernel and GP smoothing with adequate smoothing reduces NFE by 3x–5x compared to standard splines (e.g., from over 1000 to \~170 on UWaveGestureLibrary) while maintaining or slightly improving accuracy.
>
> **3\.2. Multi-View Mechanism (MV-CDE):** Adding multi-view aggregation boosts accuracy by 1%–4% over single-path smoothing and stabilizes performance by combining dynamics across multiple bandwidths, preventing drops from poorly tuned single scales.
>
> **3\.3. Convolutional Front-End (MVC-CDE):** The convolutional layer yields an additional 2%–3% accuracy gain and improves efficiency. On CharacterTrajectories, MVC GP reaches 95.07% (vs. 92.97% for MV GP) while cutting training time in half.
>
> **3\.4. GP vs. Gaussian Kernel:** GP regression provides theoretically optimal MSE reconstruction, while kernel smoothing relies on sub-optimal weights and therefore requires finer scales, taking \~2x longer to train yet yielding slightly lower accuracy.
>
> **4\. Experimental Scope** We expanded our evaluation to include several new experiments, aiming to widen the scope of our practical results.
>
> **4\.1 Multivariate Time-Series Forecasting under Extreme Missingness:** To validate our method in a standard predictive setting, we evaluated on the real-world ETTm1 dataset (24-step history to 24-step horizon). To stress-test the continuous-time framework, we simulated extreme sensor failure by dropping 30% of daily observations. Using a DLinear projection head for all models, MVC GP outperforms standard NCDE (cubic splines) and GRU baselines:
>
> **We provide our results in Table 2**: <https://bashify.io/i/YEFSK4#>
>
> **4\.2 Additional dataset (irregular setting):** We added a new [Speech Commands](https://doi.org/10.48550/arXiv.1804.03209) dataset. We applied distinct data drop rates to test scalability and evaluate performance on missing values.
>
> **We provide our results in Table 3:** <https://bashify.io/i/5ZcSyx#>
>
> MVC GP maintains accuracy on irregular data. Sparse inputs trigger more solver steps in Cubic and Linear baselines. These models require frequent function evaluations and longer training times. GP kernel smoothing provides a stable path for the solver when observations are missing. The model keeps the number of function evaluations low in these sparse settings.

---

> > ### Author Rebuttal · Reviewer_k4zT · 2026-04-01
> >
> > Thank you for your response. I will improve my score.

---

> > > ### Author Response · Authors · 2026-04-07
> > >
> > > Thank you for your positive feedback!

---

### Decision · Program_Chairs · 2026-04-30

**Decision:**

Accept (regular)

**Comment:**

The paper identifies a concrete bottleneck in Neural CDEs — the roughness of spline-interpolated control paths forcing adaptive solvers into small steps — and proposes replacing exact interpolation with kernel or Gaussian Process smoothing, with an attention-based Multi-View CDE (and its convolutional extension MVC-CDE) to recover information lost to smoothing. All four reviewers agree on the interestingness of the method, the strength of the theoretical justification, and the empirical results, with the main reservation being whether the contribution is incremental relative to discrete baselines like GRU-D and SSMs like Mamba. The rebuttal addresses these concerns substantively: head-diversity is empirically validated, the stiffest-head bottleneck is acknowledged and managed via coarse log-spaced bandwidths, a clean component-level decomposition isolates the contributions of smoothing, multi-view aggregation, and the convolutional front-end, and the scope is widened with ETTm1 forecasting under 30% missingness, Speech Commands, PEMS-SF for high-dimensional scalability, and a parameter-matched Mamba comparison. Theoretical clarifications (formal NFE definition, explicit tolerance dependence in Theorem 3.4, bias-variance-NFE trade-off in bandwidth selection) materially strengthen the paper.
Recommendation to the authors: consider engaging with Mechanistic Neural Networks also from ICML 2024; https://icml.cc/virtual/2024/poster/33046, which integrates adaptive linear solvers as efficient differentiable layers. A discussion in the camera-ready version of how MVC-CDE relates to this line of work would sharpen the positioning and suggest productive directions for follow-up.